# PerSim: Data-efficient Offline Reinforcement Learning with Heterogeneous Agents via Personalized Simulators

**Anish Agarwal**
MIT
anish90@mit.edu

**Abdullah Alomar**
MIT
aalomar@mit.edu

**Varkey Alumootil**
MIT
varkey@mit.edu

**Devavrat Shah**
MIT
devavrat@mit.edu

**Dennis Shen**
MIT
deshen@mit.edu

**Zhi Xu**
MIT
zhixu@mit.edu

**Cindy Yang**
MIT
cxy99@mit.edu

## Abstract

We consider offline reinforcement learning (RL) with heterogeneous agents under severe data scarcity, i.e., we only observe a single historical trajectory for every agent under an unknown, potentially sub-optimal policy. We find that the performance of state-of-the-art offline and model-based RL methods degrade significantly given such limited data availability, even for commonly perceived "solved" benchmark settings such as "MountainCar" and "CartPole". To address this challenge, we propose PerSim, a model-based offline RL approach which first learns a personalized simulator for each agent by collectively using the historical trajectories across all agents, prior to learning a policy. We do so by positing that the transition dynamics across agents can be represented as a latent function of latent factors associated with agents, states, and actions; subsequently, we theoretically establish that this function is well-approximated by a "low-rank" decomposition of separable agent, state, and action latent functions. This representation suggests a simple, regularized neural network architecture to effectively learn the transition dynamics per agent, even with scarce, offline data. We perform extensive experiments across several benchmark environments and RL methods. The consistent improvement of our approach, measured in terms of both state dynamics prediction and eventual reward, confirms the efficacy of our framework in leveraging limited historical data to simultaneously learn personalized policies across agents.

## 1 Introduction

Reinforcement learning (RL) coupled with expressive deep neural networks has now become a generic yet powerful solution for learning complex decision-making policies for an agent of interest; it provides the key algorithmic foundation underpinning recent successes such as game solving [34, 45, 44] and robotics [30, 22]. However, many state-of-the-art RL methods are data hungry and require the ability to query samples at will, which is infeasible for numerous settings such as healthcare, autonomous driving, and socio-economic systems. As a result, there has been a rapidly growing literature on "offline RL" [31, 24, 32, 15], which focuses on leveraging existing datasets to learn decision-making policies.

Within offline RL, we consider a regime of *severe data scarcity*: there are multiple agents and for each agent, we only observe a single historical trajectory generated under an unknown, potentially sub-optimal policy; further, the agents are *heterogeneous*, i.e., each agent has unique state transition dynamics. Importantly, the characteristics of the agents that make their transition dynamics heterogeneous are *latent*. Using such limited offline data to *simultaneously* learn a good "personalized" policy for each agent is a challenging setting that has received limited attention. Below, we use a prevalent example

35th Conference on Neural Information Processing Systems (NeurIPS 2021).

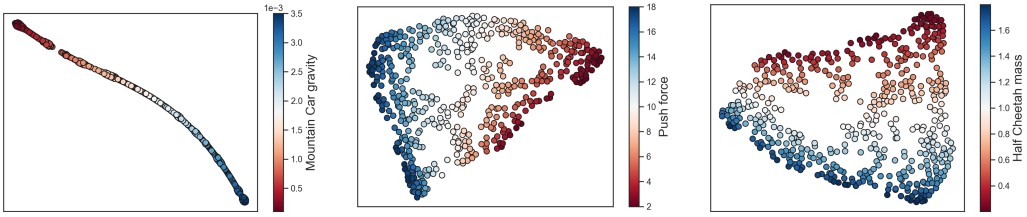

(a) MountainCar   (b) CartPole   (c) HalfCheetah

Figure 1: t-SNE visualization of the learned latent factors for the 500 heterogeneous agents. Colors indicate the value of the modified parameters in each environment (e.g., gravity in MountainCar). There is an informative low-dimensional manifold induced by the agent-specific latent factors, and there is a natural direction on the manifold along which the parameters that characterize the heterogeneity vary continuously and smoothly.

from healthcare to motivate and argue that tackling such a challenge is an important and necessary step towards building personalized decision-making engines via RL in a variety of important applications.

*A Motivating Example.* Consider a pre-existing clinical dataset of patients (agents). Our goal is to design a personalized treatment plan (policy) for each patient moving forward. Notable challenges include the following: First, each patient only provides a single trajectory of their medical history. Second, each patient is heterogeneous in that they may have a varied response for a given treatment under similar medical conditions; further, the underlying reason for this heterogeneous response across patients is likely unknown and thus not recorded in the dataset. Third, in the absence of an accurate personalized "forecasting" model for a patient's medical outcome given a treatment plan, the treatment assigned is likely to be sub-optimal. This is particularly true for complicated medical conditions like T-Cell Lymphoma. We aim to address the challenges laid out above (offline scarce data, heterogeneity, and sub-optimal policies) so as to develop a personalized "forecasting" model for each patient under any given treatment plan. Doing so will then naturally enable ideal personalized treatment policies for each patient.

*Key question.* Tackling a scenario like the one described above in a principled manner is the focus of this work. In particular, we seek to answer the following question:

> *"Can we leverage scarce, offline data collected across heterogeneous agents under unknown, sub-optimal policies to learn a personalized policy for each agent?"*

**Our Contributions.** As our main contribution, we answer this question in the affirmative by developing a structured framework to tackle this challenging yet meaningful setting. Next, we summarize the main methodological, theoretical, algorithmic, and experimental contributions in our proposed framework.

*Methodological—personalized simulators.* We propose a novel methodological framework, PerSim, to learn a policy given the data availability described above. Taking inspiration from the model-based RL literature, our approach is to first build a personalized simulator for each agent. Specifically, we use the offline data collectively available across heterogeneous agents to learn the unique transition dynamics for each agent. We do this *without* requiring access to the covariates or features that drive the heterogeneity amongst the agents. Having constructed a personalized simulator, we then learn a personalized decision-making policy separately for each agent by simply using online model predictive control (MPC) [16, 9].

*Theoretical—learning across agents.* As alluded to earlier, the challenge in building a personalized simulator for each agent is that we only have access to a single offline trajectory for any given agent. Hence, each agent likely explores a very small subset of the entire state-action space. However, by viewing the trajectories across the multitude of agents collectively, we potentially have access to a relatively larger and more diverse offline dataset that covers a much richer subset of the state-action space. Still, any approach that augments the data of an agent in this manner must address the possibly large heterogeneity amongst the agents, which is challenging as we do not observe the characteristics that make agents heterogeneous. Inspired by the literature on collaborative filtering for recommendation systems, we posit that the transition dynamics across agents can be represented as a latent function of latent factors associated with agents, states, and actions. In doing so, we establish that this function is well-approximated by a "low-rank" decomposition of separable agent, state, and action latent functions (Theorem 1). Hence, for any finite sampling of the state and action spaces, accurate model learning for each agent with offline data—generated from any policy—can be reduced to estimating a low-rank tensor corresponding to agents, states, and actions. As such, low-rank tensors can provide a useful algorithmic lens to enable model learning with offline data in RL and we hope this work leads to further research studying the relationship between these seemingly disparate fields.

*Algorithmic—regularizing via a latent factor model.* As a consequence of our low-rank representation result, we propose a natural neural network architecture that respects the constraints induced by the factorization across agents, states, and actions (Section 4). It is this principled structure, which accounts for agent heterogeneity and regularizes the model learning, that ensures the success of our approach despite access to only scarce and heterogeneous data. Further, we propose a natural extension of our framework which generalizes to unseen agents, i.e., agents that are not observed in the offline data.

*Experimental—extensive benchmarking.* Using standard environments from OpenAI Gym, we extensively benchmark PerSim against four state-of-the-art methods: a model-based online RL method (CaDM) [29], two model-free offline RL method (BCQ and CQL) [15, 25], and a model-based offline RL method (MOReL) [23]. Below, we highlight six conclusions we reach from our experiments. (i) Despite access to only a single trajectory from each agent (and no access to the covariates that drive agent heterogeneity), PerSim produces accurate personalized simulators for each agent. (ii) All benchmarking algorithms perform sub-optimally for the data availability we consider, even on simple baseline environments such as MountainCar and Cartpole, which are traditionally considered to be "solved". (iii) PerSim is able to robustly extrapolate outside the policy used to generate the offline dataset, even if the policy is highly sub-optimal (e.g., actions are sampled uniformly at random). (iv) To corroborate our latent factor representation, we find that across all environments, the learned agent-specific latent factors correspond very closely with the latent source of heterogeneity amongst agents; we re-emphasize this is despite PerSim not getting access to the agent covariates. (v) We find that augmenting the training data of an offline model-free method (e.g., BCQ) with PerSim-generated synthetic trajectories results in a significantly better average reward. (vi) As an ablation study, if we decrease the number of observed trajectories, PerSim consistently achieves a higher reward than the other baselines across most agents, indicating its robustness to data scarcity. For a visual depiction of the conclusions, see Figure 1 for the learned latent agent factors and Figure 2 for the relative prediction accuracy of the learned model using PerSim versus [29].

**Related Work.** Due to space constraints, we present a short overview of the related literature. A more detailed review is provided in Appendix B.

There are two sub-fields within RL that are of particular relevance: (i) model-based online RL and (ii) model-free offline RL. (i) In the model-based RL literature, the transition dynamics (simulator) is learnt and subsequently utilized for policy learning. These methods have been found to have far better data efficiency compared to their model-free counterparts [50, 10, 11,

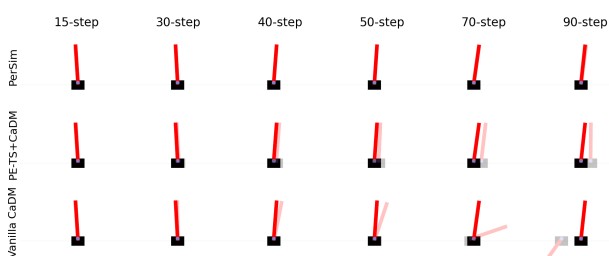

Figure 2: Visualization of prediction accuracy of the various learned models for CartPole. Actual and predicted states are denoted by the opaque and translucent objects, respectively.

26, 21, 17]. However, the current model-based literature mostly focuses on the setting where one can adaptively sample trajectories in an online manner during model-learning. A few recent works [29, 36] have considered agent heterogeneity. We compare with [29] given its strong performance in handling heterogeneous agents. (ii) In the model-free offline RL literature, one uses a pre-recorded dataset to directly learn a policy, i.e., without first learning a model. Thus far, the vast majority of offline RL methods are model-free and designed for settings that allow access to numerous trajectories from a single agent, i.e., no agent heterogeneity [15, 24, 28, 32, 51, 4, 25]. To study how much offline methods suffer if agent heterogeneity is introduced, we compare with both [15] and [25] given their strong performance with offline data. We choose these baselines as they come from a well-established literature but their abilities to simultaneously handle agent heterogeneity and sparse offline data has yet to be studied.

Model-based offline RL is still a relatively nascent field. Two recent works [23, 53] have shown that, in certain settings, model-based offline methods can outperform their model-free counterparts on benchmark environments. In [53], authors exhibit this using existing online model-based methods with minimal changes. However, both works restrict attention to the setting where there is only one agent (or environment) and a large number of observations from that agent are available. Hence, we study how these methods perform when given only sparse data from heterogeneous agents, and we find their performance does suffer. Extending these model-based offline RL methods to work with sparse data from heterogeneous agents, possibly by building upon the latent low-rank functional representation we propose, remains an interesting future work.

## 2 Problem Statement

We consider the standard RL framework with $N$ heterogeneous agents. We index agents with $n \in [N]$[1]. Formally, we describe our problem as a Markov Decision Process (MDP) defined by the tuple $(\mathcal{S}, \mathcal{A}, P_n, R_n, \gamma_n, \mu_n)$. Here, $\mathcal{S}$ and $\mathcal{A}$ denote the state and action spaces, respectively, which are common across agents. For every agent $n$, $P_n(s'|s,a)$ is the unknown transition kernel, $R_n(s,a)$ is the immediate reward received, $\gamma_n \in (0,1)$ is the discounting factor, and $\mu_n$ is the initial state distribution.

**Observations.** We consider an offline RL setting where we observe a single trajectory of length $T$ for each of the $N$ heterogeneous agents. Formally, for each agent $n$ and time step $t$, let $s_n(t) \in \mathcal{S}$, $a_n(t) \in \mathcal{A}$, and $r_n(t) \in \mathbb{R}$ denote the observed state, action, and reward. We denote our observations as $\mathcal{D} = \{(s_n(t), a_n(t), r_n(t)) : n \in [N], t \in [T]\}$.

**Goals.** We state our two primary goals below.

*"Personalized" trajectory prediction.* For a given agent $n$ and state-action pair $(s,a) \in \mathcal{S} \times \mathcal{A}$, we would like to estimate $\mathbb{E}[s'_n | (s_n, a_n) = (s,a)]$, i.e., given the observations $\mathcal{D}$, we would like to build a "personalized" simulator (i.e., a model of the transition dynamic) for each agent $n$.

*"Personalized" model-based policy learning.* To test the efficacy of the personalized simulator, we would like to subsequently use it to learn a good decision-making policy for agent $n$, denoted as $\pi_n : \mathcal{S} \to \mathcal{A}$, which takes as input a given state and produces a corresponding action.

*A causal inference lens.* We note that this problem can be thought of as a counterfactual prediction problem. Our goal is to build a personalized simulator that answers the following question: "what would have happened had the agent took a sequence of actions other than the one we observe in the dataset?". Broadly speaking, our aim in PerSim is to answer such questions by observing how other agents behave under varied sequences of actions. More generally, we hope PerSim serves to further link causal inference with offline RL, which is fundamentally regarded as a counterfactual inference problem [31].

## 3 Latent Low-rank Factor Representation

To address the goals, we introduce a latent factor model for the transition dynamics. Leveraging latent factors have been successful in recommendation systems for overcoming heterogeneity of users. Such models have also been shown to provide a "universal" representation for multi-dimensional exchangeable arrays [5, 18]. Indeed, our latent model holds for known environments such as MountainCar.

Assume $\mathcal{S} \subseteq \mathbb{R}^D$, i.e., the state is $D$-dimensional. Let $s_{nd}$ refer to the $d$-th coordinate of $s_n$. We posit the transition dynamics (in expectation) obey the following model: for every agent $n$ and state-action pair $(s,a)$,

$$\mathbb{E}[s'_{nd} | (s_n, a_n) = (s,a)] = f_d(\theta_n, \rho_s, \omega_a), \tag{1}$$

where $s'_{nd}$ denotes the $d$-th state coordinate after taking action $a$. Here, $\theta_n \in \mathbb{R}^{d_1}$, $\rho_s \in \mathbb{R}^{d_2}$, $\omega_a \in \mathbb{R}^{d_3}$ for some $d_1, d_2, d_3 \geq 1$ are latent feature vectors capturing relevant information specific to the agent, state, and action; $f_d : \mathbb{R}^{d_1} \times \mathbb{R}^{d_2} \times \mathbb{R}^{d_3} \to \mathbb{R}$ is a latent function capturing the model relationship between these latent feature vectors. We assume $f_d$ is $L$-Lipschitz and the latent features are bounded.

**Assumption 1.** Suppose $\theta_n \in [0,1]^{d_1}, \rho_s \in [0,1]^{d_2}, \omega_a \in [0,1]^{d_3}$, and $f_d$ is $L$-Lipschitz with respect to its arguments, i.e., $|f_d(\theta_{n'}, \rho_{s'}, \omega_{a'}) - f_d(\theta_n, \rho_s, \omega_a)| \leq L(\|\theta_{n'} - \theta_n\|_2 + \|\rho_{s'} - \rho_s\|_2 + \|\omega_{a'} - \omega_a\|_2)$.

For notational convenience, let $\tilde{f}_d : [N] \times \mathcal{S} \times \mathcal{A} \to \mathbb{R}$ be such that $\tilde{f}_d(n,s,a) = \mathbb{E}[s'_{nd} | (s_n, a_n) = (s,a)]$.

**Theorem 1.** Suppose Assumption 1 holds and without loss of generality, let $d_1, d_3 \leq d_2$. Then for all $d \in [D]$ and any $\delta > 0$, there exists $h_d : [N] \times \mathcal{S} \times \mathcal{A} \to \mathbb{R}$, such that $h_d(n,s,a) = \sum_{\ell=1}^r u_\ell(n) v_\ell(s,d) w_\ell(a)$ with $r \leq C\delta^{-(d_1+d_3)}$ and $\|h_d - \tilde{f}_d\|_\infty \leq 2L\delta$, where $C$ is an absolute constant.

Theorem 1 suggests that under the model in (1), the transition dynamics are well approximated by a low-rank order-three functional tensor representation. In fact, as we show below, for a classical non-linear dynamical system, it is exact.

*An example.* We show that the MountainCar transition dynamics [8] exactly satisfies this low-rank tensor representation. In MountainCar, the state $s_n = [s_{n1}, s_{n2}]$ consists of car (agent) $n$'s position and velocity, i.e., $\mathcal{S} \subseteq \mathbb{R}^2$; the action $a_n$ is a scalar that represents the applied acceleration, i.e., $\mathcal{A} \subseteq \mathbb{R}$. For

---

[1] For any positive integer $N$, let $[N] = \{1,...,N\}$.

car $n$, parameterized by gravity $g_n$, the (deterministic) transition dynamics given action $a_n$ are

$$s'_{n1} = s_{n1} + s_{n2} - \frac{g_n \cos(3s_{n1})}{2} + \frac{a_n}{2}, \quad s'_{n2} = s_{n2} - g_n \cos(3s_{n1}) + a_n.$$

**Proposition 1.** In MountainCar, $r = 3$.

**Model Learning & Tensor Estimation.** Consider any finite sampling of the states $\tilde{\mathcal{S}} \subset \mathcal{S}$ and actions $\tilde{\mathcal{A}} \subset \mathcal{A}$. Let $\mathcal{X} = [X_{nsad}] \in \mathbb{R}^{N \times |\tilde{\mathcal{S}}| \times |\tilde{\mathcal{A}}| \times D}$ be the order-four tensor, where $X_{nsad} = f_d(\theta_n, \rho_s, \omega_a)$. Hence, to learn the model of transition dynamics for all the agents over $\tilde{\mathcal{S}}$, $\tilde{\mathcal{A}}$, it is sufficient to estimate the tensor, $\mathcal{X}$, from observed data. The offline data collected for a given policy induces a corresponding observation pattern of this tensor. Whether the complete tensor is recoverable, is determined by this induced sparsity pattern and the rank of $\mathcal{X}$. Notably, Theorem 1 suggests that $\mathcal{X}$ is low-rank under mild regularity conditions. Therefore, the question of model identification, i.e., completing the tensor $\mathcal{X}$, boils down to conditions on the offline data in terms of the observation pattern that it induces in the tensor. In the existing tensor estimation literature, there are few sparsity patterns for which the underlying tensor can be provably recovered, provided $\mathcal{X}$ has a low-rank structure. They include: (i) each entry of the tensor is observed independently at random with sufficiently high-probability [6, 35, 43]; (ii) the entries that are observed are *block-structured* [2, 42]. However, the most general set of conditions on the sparsity pattern under which the underlying tensor can be faithfully estimated, i.e., the model is identified for our setup, remains an important and active area of research.

## 4   PerSim Algorithm

We now detail our proposed algorithm which is composed of two steps: (i) build a personalized simulator for each agent $n$ given offline observations $\mathcal{D}$, which is comprised of a single trajectory per agent; (ii) learn a personalized decision-making policy using MPC.

**Step 1. Learning Personalized Simulators.** Theorem 1 suggests that the transition dynamics can be represented as a low-rank tensor function with latent functions associated with the agents, states, and actions. This guides the design of a simple, regularized neural network architecture: we use three separate neural networks to learn the agent, state,

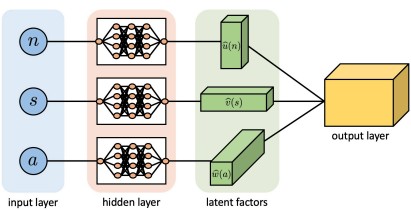

Figure 3: Neural Network Architecture

and action latent functions, i.e., we remove any edges between these three neural networks. See Figure 1 for a visual depiction of our proposed architecture. Specifically, to estimate the next state for a given agent $n$ and state-action pair $(s, a) \in \mathcal{S} \times \mathcal{A}$, we learn the following functions:

1. An agent encoder $g_u : [N] \to \mathbb{R}^r$, parameterized by $\psi$, which estimates the latent function associated with an agent, i.e., $\widehat{u}(n) = g_u(n; \psi)$.
2. A state encoder $g_v : \mathcal{S} \to \mathbb{R}^{D \times r}$ parameterized by $\phi$, which estimates $D$ latent functions, where each vector is associated with the corresponding state coordinate, i.e., $\widehat{v}(s) := (\widehat{v}(s, 1), ..., \widehat{v}(s, D))^T = g_v(s; \phi)$.
3. An action encoder $g_w : \mathcal{A} \to \mathbb{R}^r$ parameterized by $\theta$, which estimates the latent function associated with the action, i.e., $\widehat{w}(a) = g_w(a; \theta)$.

Then, our estimate of the expected $d$-th coordinate of the next state is given by $\widehat{\mathbb{E}}[s'_{nd} | (s_n, a_n) = (s, a)] = \sum_{\ell=1}^{r} \widehat{u}_\ell(n) \widehat{v}_\ell(s, d) \widehat{w}_\ell(a)$. We optimize our agent, state, and action encoders by minimizing the squared loss: $\mathcal{L}(s, a, n, s'; \psi, \phi, \theta) = \sum_{d=1}^{D} \left( s'_{nd} - \sum_{\ell=1}^{r} \widehat{u}_\ell(n) \widehat{v}_\ell(s, d) \widehat{w}_\ell(a) \right)^2$.

**Step 2. Learning a Decision-making Policy.** We use MPC to select the next action, as common practice in the literature [29]. Since the offline data may not span the entire state-action space, planning using a learned simulator without any regularization may result in "model exploitation" [31]. Thus, when choosing the action via MPC, we choose the first action from the sequence of actions with the best *average* reward across an ensemble of $M$ simulators. In our experiments, we find that the simple technique of using the state difference instead of the raw state improves performance. For a detailed description of the algorithm, including the pseudo-code, please refer to Appendix D.

# 5 Experiments

In this section, through a systematic collection of experiments on a variety of benchmark environments, we demonstrate that PerSim consistently outperforms state-of-the-art model-based and offline model-free RL algorithms, in terms of prediction error and reward, for the data regime we consider.

## 5.1 Setup and Benchmarks

We evaluate PerSim on three benchmark environments from OpenAI Gym [8]: MountainCar, CartPole, and HalfCheetah. A detailed description of each environment can be found in Appendix E.

**Heterogeneous Agents.** We introduce agent heterogeneity across the various environments by modifying the covariates that characterize the transition dynamics of an agent. This is in line with what has been done in the literature [29] to study algorithmic robustness to agent heterogeneity. The range of parameters we consider for each of the three environments is given in Table 4 of Appendix E; for example, we create heterogeneous agents in MountainCar by varying the gravity parameter of a car (agent) within the interval $[0.01, 0.035]$.

For each environment, we uniformly sample 500 covariates (i.e., heterogeneous agents) from the parameter ranges displayed in Table 4 of Appendix E. We then select five of the 500 agents to be our "test" agents, i.e., these are the agents for which we report the prediction error and eventual reward for the various RL algorithms. Our rationale for selecting these five agents is as follows: one of the five is the default covariate parameter in an environment; the other four are selected so as to cover the "extremes" of the parameter range. Due to space constraints, we show results for three test agents in this section; the results for the remaining two agents can be found in Appendix F. We note that the conclusions we draw from our experiments continue to hold over all test agents we evaluate on.

**Offline Data from Sub-optimal Policies.** To study how robust the various RL algorithms are to the "optimality" of the sampling policy used to generate the historical trajectories, we create four offline datasets of 500 trajectories (one per agent) for each environment as follows:

*(i) Pure policy*. For each agent, actions are sampled according to a fixed policy that has been trained to achieve "good" performance. Specifically, for each environment, we first train a policy in an online fashion for a few training agents (see Table 4 for details). We pick the training agents to be uniformly spread throughout the training range to ensure reasonable performance for all agents. See Appendix E for details about the average reward achieved across all agents using this procedure. For MountainCar and CartPole, we use DQN [34] to train the sampling policy; for HalfCheetah, we use TD3 [14]. The policies are trained to achieve rewards of approximately -200, 120, and 3000 for MountainCar, CartPole, and HalfCheetah, respectively. Then for each of the 500 agents, we sample one trajectory using the policy trained on the training agent with the closest parameter value. *(ii) Random*. Actions are selected uniformly at random. *(iii/iv) Pure-$\varepsilon$-20/Pure-$\varepsilon$-40*. For Pure-$\varepsilon$-20/40, actions are selected uniformly at random with probability 0.2/0.4, respectively, and selected via the pure policy otherwise.

The "pure policy" dataset has relatively optimal transition dynamics for each agent compared to the other three policies. This is likely the ideal scenario when we only have access to limited offline data. However, such an ideal sampling procedure is hardly met in practice. Real-world data often contains at least some amount of "trial and error" in terms of how actions are chosen; we model this by sampling a fraction of the trajectory at random.

**Benchmarking Algorithms.** We compare with four state-of-the-art RL algorithms; (i) one from the online model-based literature; (ii) two from the offline model-free literature; (iii) and one offline model-based approach. In Appendix D, we give implementation details.

*Vanilla CaDM + PE-TS CaDM [29]*. As aforementioned, we choose CaDM as a baseline given its superior performance against other popular model-based (and meta-learning) methods in handling heterogeneous agents. CaDM tackles heterogeneity by learning a context vector using the recent trajectory of a given agent, with a common context encoder across all agents. In our evaluation, we compare against two CaDM variants discussed in [29] with respect to both model prediction error and eventual reward.

*BCQ-P + BCQ-A [15]*. BCQ is an offline model-free RL method that has been shown to exhibit excellent performance in the standard offline setting. In light of this, we consider two BCQ baselines: (i) BCQ-Population (or BCQ-P), where a *single* policy is trained using data from all available (heterogeneous) agents, i.e., all 500 observed trajectories; (ii) BCQ-Agent (or BCQ-A), where a separate policy is learned for each of the test agents using just the single observed trajectory associated with that test agent. We compare PerSim against both BCQ variants with respect to the eventual reward in order to study the effect that data scarcity and agent heterogeneity can have on standard offline RL methods.

Table 1: Prediction error.

| Data | Method | MountainCar | | | CartPole | | | HalfCheetah | | |
|---|---|---|---|---|---|---|---|---|---|---|
| | | 0.01 | 0.025 | 0.035 | (2/0.5) | (10/0.85) | (10/0.15) | (0.3/1.7) | (1.7/0.3) | (0.3/0.3) |
| Pure | PerSim | 0.025 (0.97) | 0.014 (0.94) | 0.039 (0.80) | 0.01 (1.00) | 0.01 (1.00) | 0.035 (1.00) | 1.194 (0.92) | 4.064 (0.67) | 4.070 (0.81) |
| | Vanilla CaDM | 0.149 (0.74) | 0.177 (-18.4) | 0.238 (-30.1) | 0.403 (0.71) | 0.039 (0.98) | 2.531 (-0.53) | 3.902 (0.47) | 3.851(0.49) | 6.308 (0.38) |
| | PE-TS CaDM | 0.326 (-1.91) | 0.154 (-1.11) | 0.148 (-0.18) | 0.031 (0.99) | 0.06 (1.00) | 0.319 (0.65) | 3.147 (0.61) | 3.080 (0.68) | 5.270 (0.57) |
| Random | PerSim | 0.04 (1.00) | 0.01 (1.00) | 0.01 (1.00) | 0.014 (0.98) | 0.06 (1.00) | 0.152 (0.88) | 1.194 (0.92) | 4.064 (0.67) | 4.070 (0.81) |
| | Vanilla CaDM | 0.256 (0.43) | 0.134 (0.71) | 0.217 (-1.77) | 0.172 (0.10) | 0.098 (0.64) | 3.307 (-0.73) | 4.030 (0.44) | 4.121 (0.43) | 4.446 (0.67) |
| | PE-TS CaDM | 0.242 (0.31) | 0.101 (0.87) | 0.075 (0.97) | 0.564 (-1.36) | 0.216 (0.45) | 3.764 (-1.10) | 2.735 (0.73) | 2.756 (0.69) | 4.141 (0.77) |
| Pure-$\varepsilon$-20 | PerSim | 0.04 (1.00) | 0.02 (1.00) | 0.04 (1.00) | 0.0 (1.00) | 0.01 (1.00) | 0.048 (0.98) | 1.172 (0.92) | 4.283 (0.66) | 3.832 (0.84) |
| | Vanilla CaDM | 0.227 (0.31) | 0.101 (-0.37) | 0.157 (-2.25) | 0.270 (-0.98) | 0.058 (0.94) | 2.193 (-0.43) | 3.613 (0.53) | 3.455 (0.54) | 6.046 (0.48) |
| | PE-TS CaDM | 0.350 (-2.57) | 0.139 (0.75) | 0.130 (0.89) | 0.639 (-1.21) | 0.148 (0.38) | 2.680 (-0.56) | 2.913 (0.70) | 2.959 (0.65) | 4.818 (0.65) |
| Pure-$\varepsilon$-40 | PerSim | 0.06 (1.00) | 0.06 (0.99) | 0.03 (1.00) | 0.0 (1.00) | 0.0 (1.00) | 0.018 (1.00) | 1.016 (0.94) | 4.021 (0.66) | 3.742 (0.85) |
| | Vanilla CaDM | 0.199 (0.54) | 0.119 (0.38) | 0.187 (-9.00) | 0.233 (-0.88) | 0.035 (0.99) | 3.286 (-0.62) | 3.685 (0.49) | 3.612 (0.48) | 6.000 (0.39) |
| | PE-TS CaDM | 0.639 (-2.27) | 0.192 (-6.06) | 0.157 (0.55) | 0.051 (0.98) | 0.010 (1.00) | 0.411 (0.55) | 3.021 (0.67) | 3.025 (0.61) | 5.075 (0.62) |

*CQL-P + CQL-A* [25]. CQL is considered one of the state-of-the-art offline model-free RL methods. Similar to BCQ, we consider two CQL baselines: CQL-P and CQL-A, which are defined analogously to BCQ-P and BCQ-A.

*MOReL-P + MOReL-A* [23]. MOReL is a recent work that proposes a model-based approach for offline RL. MOReL does not consider a setting where data is collected from heterogeneous agents, unlike CaDM and PerSim. However, we include MOReL as a baseline to provide a more comprehensive evaluation. Similar to BCQ and CQL, we consider two MOReL baselines: MOReL-P and MOReL-A, which are defined analogously to BCQ-P and BCQ-A.

### 5.2 Model Learning: Prediction Error

The core of PerSim is to build a personalized simulator for each agent. This is also the case for two of the methods we compare against, Vanilla CaDM and PE-TS CaDM. Thus, for each of these algorithms, we first evaluate the accuracy of the learned transition dynamics for each agent, focusing on long-horizon model prediction. Specifically, given an initial state and an unseen sequence of 50 actions, the task is to predict the next 50-step state trajectory for the test agents. The sequence of 50 actions are chosen according to an unseen test policy that is different than the policies used to sample the training dataset. Precisely , the test policies are fitted via DQN for MountainCar and CartPole and via TD3 for HalfCheetah for the agent with the default covariate parameters. The test policies were such that they achieved rewards of -150, 150, and 4000 for MountainCar, CartPole and HalfCheetah, respectively.

For each environment and for PerSim, Vanilla CaDM, and PE-TS CaDM, we train four separate models using each of the four offline datasets described earlier. We repeat the process 200 times (totaling 200 trajectories for each test agent) and report the *mean* root-mean-squared error (RMSE) over all 50-steps and over all 200 trajectories. In addition to RMSE, we provide the *median* $R^2$ (in parentheses within the tables) to facilitate a better comparison in terms of relative error. The results are summarized in Table 1 for MountainCar, CartPole, and HalfCheetah. Further experimental results regarding prediction error can be found in Appendix F.1.

**PerSim Accurately Learns Personalized Simulators.** Consistently across environments, PerSim outperforms both Vanilla CaDM and PE-TS CaDM for most test agents. RMSE for PerSim is notably lower by orders of magnitude in MountainCar and CartPole. Indeed, in these two environments, $R^2$ for PerSim is nearly one (i.e., the maximum achievable) across most agents, while for the two CaDM variants it is notably lower. In a number of experiments, the two CaDM variants have negative $R^2$ values, i.e., their predictions are worse than simply predicting the average true state across the test trajectory. For HalfCheetah, which has a relatively high-dimensional state space (it is 18-dimensional), PerSim continues to deliver reasonably good predictions for each agent despite the challenging data availability. Though PerSim still outperforms CaDM in HalfCheetah, we note CaDM's performance is more comparable in this environment.

For a more visual representation of PerSim's prediction accuracy, refer back to Figure 2 in Section 1, which shows the predicted and actual state for different horizon lengths in CartPole; there, we see that PerSim consistently produces state predictions that closely match the actual state observed from the true environment. Additional visualizations across environments are provided in Appendix F.1.

**Rethinking Model-based RL for Our Setting.** Altogether, these results indicate that existing model-based RL methods cannot effectively learn a model of the transition dynamics simultaneously across

*all* heterogeneous agents (e.g., CaDM sometimes has negative $R^2$ values) if only given access to sparse offline data. It has been exhibited [19, 53, 31] that certain model-based methods that were originally targeted for the online setting can potentially still deliver reasonable performance with offline data and minimal algorithmic change. Our experiments offer evidence that model-based RL methods, even those which are optimized to work with heterogeneous agents, do not provide a uniformly good "plug-in" solution for the particular data availability we consider. In contrast, our latent factor approach consistently learns a reasonably good model of the dynamics across all agents.

**Latent Factors Capture Agent Heterogeneity.** The poor performance of standard model-based RL methods for the data availability setting we consider emphasizes the need for new principled approaches. Ours is one such approach, where we posit a low-rank latent factor representation of the agents, states, and actions. To further validate our approach, we visualize the learned latent agent factors associated with the 500 heterogeneous agents in each environment in Figure 1 of Section 1. Pleasingly, across environments, the latent factors correspond closely with the heterogeneity across agents.

## 5.3 Policy Performance: Average Reward

In this section, we evaluate the average reward achieved by PerSim compared to several state-of-the-art model-based and model-free offline RL methods. For the PerSim and the two CaDM variants, we follow [29] in utilizing MPC to make policy decisions on top of the learned model. We include an additional baseline, "True env + MPC", where we apply MPC on top of the actual ground-truth environment; this allows us to quantify the difference in reward when using MPC with the actual environment versus the learned simulators. For the model-free methods (e.g., BCQ and CQL variants), we can directly apply the policy resulted from the learned $Q$-value. Likewise, we directly apply the policy trained by MOReL. We use the trained policy for this model-based method instead of utilizing MPC to faithfully follow the method described in [23].

We evaluate the performance of each method using the average reward over 20 episodes for model-based methods and the average over 100 episodes for model-free methods. We repeat each experiment five times and report the mean and standard deviation. Table 2 presents the results for MountainCar, CartPole, and HalfCheetah. Further experimental results regarding reward are given in Appendix F.2.

As a high-level summary, in most experiments, *PerSim either achieves the best reward or close to it.* This not only reaffirms our prediction results in Section 5.2, but also is particularly encouraging for PerSim since the policy utilized is simply MPC and not optimized as done in model-free approaches and in MOReL. Furthermore, these results corroborate the appropriateness of our low-rank latent factor representation and our overall methodological framework as a principled solution to this challenging yet meaningful setting within offline RL. In what follows, we highlight additional interesting conclusions.

**"Solved" Environments are Not Actually Solved.** MountainCar and CartPole are commonly perceived as simple, "solved" environments within the RL literature. Yet, results in Table 2 demonstrate that the offline setting with scarce and heterogeneous data impose unique challenges, and undoubtedly warrants a new methodology. We find state-of-the-art model-based and model-free methods perform poorly on some of the test agents in MountainCar and CartPole. In comparison, PerSim's performance in these environments is close to that of MPC planning using the ground-truth environment across all test agents, thereby confirming the success of learning the personalized simulators in Step 1 of our algorithm. In certain rare cases (e.g., MountainCar test agent 0.01) where other baselines have a comparable performance to PerSim, we see that these baselines even outperforms True env + MPC. This indicates that the bottleneck in these experiments is using MPC for policy planning rather than the learned simulator in PerSim.

**PerSim Robustly Extrapolates with Sub-optimal Data.** Crucially, across all environments and offline data generating processes, PerSim remains the most *consistent and robust* winner. This is a much desired property for offline RL. In real-world applications, the policy used to generate the offline dataset is likely unknown. Thus, for broad applicability of a RL methodological framework, it is vital that it is robust to sub-optimality in how the dataset was generated, e.g., the dataset may contain a significant amount of "trial and error" (i.e, randomized actions). Indeed, this is one of the primary motivations to use RL in such settings in the first place.

As mentioned earlier, the four offline datasets correspond to varying degrees of "optimality" of the policy used to sample agent trajectories. We highlight that PerSim achieves uniformly good reward, even with random data, i.e., the offline trajectories are produced using totally random actions. This showcases that by first learning a personalized simulator for each agent, PerSim is able to robustly *extrapolate* outside the policy used to generate the data, however sub-optimal that policy might be. In contrast, BCQ and CQL are not robust to such sub-optimality in the offline data generation. For example, for HalfCheetah, BCQ achieves reasonable performance primarily when trained on "optimal"

Table 2: Average reward.

| Data | Method | MountainCar | | | CartPole | | | HalfCheetah | | |
|---|---|---|---|---|---|---|---|---|---|---|
| | | 0.01 | 0.025 | 0.035 | (2/0.5) | (10/0.85) | (10/0.15) | (0.3/1.7) | (1.7/0.3) | (0.3/0.3) |
| Pure | PerSim | -56.80±1.83 | -189.4±6.44 | -210.6±4.27 | 199.7±0.58 | 193.8±4.28 | 192.0±2.28 | 1984 ±763 | 997.0±403 | 714.7±314 |
| | Vanilla CaDM | -106.3±44.1 | -432.3±117 | -471.8±43.0 | 168.0±19.7 | 190.8±6.80 | 58.10±10.7 | 50.31±71.7 | -134.0±81.1 | 11.39±171 |
| | PE-TS CaDM | -74.23±16.5 | -492.3±13.3 | -500.0±0.0 | 92.30±44.8 | 193.6±8.30 | 127.5±9.50 | 481.1±252 | 503.7±181 | 553.0±127 |
| | BCQ-P | -67.60±22.3 | -267.8±202 | -295.1±180 | 166.2±39.3 | 181.2±13.5 | 182.8±15.0 | 549.8±322 | 2006 ±153 | -65.18±92.8 |
| | BCQ-A | -44.79±0.08 | -380.7±170 | -500.0±0.0 | 65.40±67.5 | 79.20±69.5 | 132.1±85.0 | -262.7±96.6 | -139.0±236 | 165.6±83.1 |
| | CQL-P | -176.1±45.2 | -316.4±26.4 | -362.9±17.9 | 154.4±17.7 | 190.9±0.7 | 170.2±0.6 | -353.5±78.4 | -453.6±71.9 | -476.7±129 |
| | CQL-A | -44.60±0.0 | -500.0±0.0 | -499.3±0.7 | 122.8±63.4 | 179.4±20.6 | 193.6±5.4 | -65.00±105 | -257.9±35.9 | -279.6±34.4 |
| | MOReL-P | -46.00±1.1 | -500.0±0.0 | -500.0±0.0 | 35.80±1.4 | 96.60±24.1 | 66.40±24.5 | -1297 ±519 | -1256 ±627 | -1470 ±727 |
| | MOReL-A | -373.0±33.5 | -500.0±0.0 | -500.0±0.0 | 33.70±3.8 | 27.50±0.1 | 10.10±0.7 | -726.2±4.9 | -666.1±42.6 | -841.7±39.5 |
| Random | PerSim | -57.70±5.63 | -186.6±4.25 | -210.1±4.48 | 197.7±7.82 | 193.0±6.60 | 185.7± 3.49 | 2124 ±518 | 2060 ±900 | 472.0±56.9 |
| | Vanilla CaDM | -62.57±5.11 | -479.3±21.7 | -497.5±4.39 | 150.5±15.7 | 175.6±5.80 | 65.10±16.3 | 288.4±32.4 | 362.9±55.4 | 351.8±34.9 |
| | PE-TS CaDM | -82.00±3.47 | -500.0±0.0 | -500.0±0.0 | 88.60±18.5 | 196.1±3.00 | 171.0±21.7 | 754.6±242 | 744.5±281 | 767.4±214 |
| | BCQ-P | -500.0±0.0 | -500.0±0.0 | -500.0±0.0 | 44.80±34.0 | 57.91±53.0 | 36.40±36.0 | -1.460±0.16 | -1.750±0.22 | -1.690±0.19 |
| | BCQ-A | -500.0±0.0 | -500.0±0.0 | -500.0±0.0 | 43.90±16.4 | 21.10±5.81 | 39.50±12.1 | -498.9±108 | -113.3±13.0 | -159.5±51.7 |
| | CQL-P | -500.0±0.0 | -500.0±0.0 | -500.0±0.0 | 39.90±29.9 | 77.90±26.2 | 148.7±17.1 | -481.5±25.9 | -442.2±56.4 | -672.4±17.6 |
| | CQL-A | -500.0±0.0 | -500.0±0.0 | -500.0±0.0 | 67.40±49.1 | 30.60±26.4 | 7.000±1.9 | -0.700±0.4 | -2.800±0.8 | -0.600±0.3 |
| | MOReL-P | -45.00±0.3 | -500.0±0.0 | -500.0±0.0 | 50.20±7.9 | 68.90±17.4 | 40.40±12.4 | -102.3±45.7 | -188.7±37.1 | -181.0±67.7 |
| | MOReL-A | -495.0±3.9 | -500.0±0.0 | -500.0±0.0 | 35.80±0.4 | 27.50±0.7 | 10.60±0.2 | -430.8±195 | -673.7±39.3 | -365.5±97.4 |
| Pure-ε-20 | PerSim | -54.20±0.56 | -191.2±6.70 | -199.7±3.99 | 199.8±0.24 | 199.1±1.30 | 197.8±1.68 | 3186 ±604 | 1032±232 | 1121±243 |
| | Vanilla CaDM | -56.73±4.20 | -463.2±57.5 | -478.9±35.8 | 171.1±38.1 | 193.4±2.10 | 64.20±10.0 | 412.0±152 | 31.92±109 | 460.2±159 |
| | PE-TS CaDM | -107.6±36.3 | -500.0±0.0 | -500.0±0.0 | 98.30±42.9 | 198.6±0.40 | 141.1±12.0 | 1082 ±126 | 1125 ±132 | 1067 ±64.3 |
| | BCQ-P | -71.21±24.4 | -286.6±196 | -328.3±158 | 98.90±30.2 | 161.1±15.5 | 86.10±72.1 | 254.6±352 | 406.7±71.1 | 385.9±57.1 |
| | BCQ-A | -364.5±180 | -260.6±51.4 | -204.5±68.9 | 67.30±62.2 | 65.60±51.8 | 140.0±80.6 | 376.8±102 | 84.66±53.3 | 230.1±10.0 |
| | CQL-P | -79.40±16.1 | -357.9±12.5 | -407.6±15.3 | 163.7±13.6 | 198.1±2.6 | 190.4±6.6 | 838.7±24.5 | 3155 ±125 | 539.9±313 |
| | CQL-A | -44.70±0.1 | -500.0±0.0 | -500.0±0.0 | 42.40±11.9 | 199.0±1.9 | 199.8±0.2 | -15.50±9.0 | -73.00±26.3 | -108.0±64.6 |
| | MOReL-P | -83.50±15.6 | -357.0±13.8 | -407.4±17.1 | 166.7±13.6 | 197.7±2.7 | 189.4±7.1 | 0.600±209 | -171.2±125 | -219.9±76.7 |
| | MOReL-A | -44.60±0.1 | -500.0±0.0 | -500.0±0.0 | 41.40±13.1 | 198.8±2.1 | 199.9±0.2 | -781.0±37.9 | -613.1±49.9 | -847.1±64.7 |
| Pure-ε-40 | PerSim | -54.60±0.55 | -189.7±7.14 | -200.3±2.26 | 199.9±0.18 | 198.0±1.21 | 197.4±1.72 | 2590 ±813 | 1016 ±283 | 1365 ±582 |
| | Vanilla CaDM | -55.23±0.76 | -481.7±25.3 | -496.2±4.31 | 160.6±46.6 | 191.9±6.40 | 79.60±31.6 | 465.6±49.2 | 452.7±130 | 720.0±74.9 |
| | PE-TS CaDM | -102.3±20.3 | -500.0±0.0 | -500.0±0.0 | 91.90±67.6 | 197.0±1.40 | 143.5±17.5 | 1500 ±246 | 1218 ±221 | 1339 ±54.8 |
| | BCQ-P | -50.01±7.50 | -373.6±180 | -352.0±211 | 28.90±6.80 | 31.80±25.9 | 18.50±11.1 | 78.25±200 | 173.8±189 | 417.1±155 |
| | BCQ-A | -94.87±0.88 | -358.7±201 | -486.5±20.6 | 34.60±1.55 | 47.71±48.7 | 23.20±9.44 | 269.2±60.7 | -181.5±57.4 | 193.0±31.8 |
| | CQL-P | -61.40±2.1 | -366.9±21.3 | -429.1±18.8 | 182.2±18.0 | 198.3±2.9 | 191.9±8.7 | 808.5±240 | 1662 ±221 | -156.3±119 |
| | CQL-A | -44.70±0.0 | -500.0±0.0 | -490.1±9.6 | 20.70±0.7 | 134.2±10.2 | 9.700±8.1 | -6.200±2.9 | -386.0±42.4 | 37.10±154 |
| | MOReL-P | -61.30±2.3 | -373.2±19.3 | -428.5±21.0 | 178.9±18.7 | 197.9±3.1 | 190.6±9.3 | 8.500±61.6 | -114.2±72.2 | -195.9±77.3 |
| | MOReL-A | -44.70±0.0 | -500.0±0.0 | -492.1±9.3 | 20.70±0.8 | 135.4±11.2 | 10.80±8.8 | -325.9±17.1 | -644.5±18.8 | -798.8±130 |
| | True env+MPC | -53.95±4.10 | -182.9±22.9 | -197.5±20.7 | 200.0±0.0 | 198.4±7.20 | 200.0±0.0 | 7459 ±171 | 42893±6959 | 66675±9364 |

offline data (i.e., pure policy). This is because BCQ, by design, is conservative and is regularized to only pick actions that are close to what is seen in the offline data. In summary, PerSim's ability to successfully extrapolate, even with sub-optimal offline data, makes it a suitable candidate for real-world applications.

## 5.4 Combination with Model-Free Methods: PerSim+BCQ/CQL

We explore whether simulated trajectories produced from PerSim can be used to improve the performance of model-free RL methods such as BCQ and CQL. In particular, instead of using a single observed trajectory to learn an agent-specific policy, as is done in BCQ-A and CQL-A, we use PerSim to generate 5 "synthetic" trajectories for that agent. We then use both the synthetic and the observed trajectories to train both BCQ and CQL (denoted by PerSim-BCQ-5 and PerSim-CQL-5, respectively). Note that using model-based methods to augment the data used by a model-free method has been explored in the literature [46, 38, 19].

If the learned model is accurate, improvements for BCQ and CQL are naturally anticipated. Table 3 confirms that this is indeed the case for PerSim. Across all environments, augmenting the training data with PerSim results in a significantly better average reward for most test agents. Specifically, the performance of PerSim-BCQ-5 and PerSim-CQL-5 indicates that a personalized BCQ/CQL policy trained on few PerSim-generated trajectories is superior to: (i) a single BCQ/CQL policy trained using

all 500 trajectories (e.g., BCQ-P); and (ii) a personalized BCQ/CQL policy trained using a single observed trajectory (e.g., BCQ-A). Refer to Appendix F.4 for more details about the experiment.

Table 3: BCQ+PerSim

| Environment | Method | 0.01 | 0.05 | 0.01 | 0.025 | 0.035 |
|---|---|---|---|---|---|---|
| MountainCar | BCQ-P | -500.0±0.0 | -500.0±0.0 | -500.0±0.0 | -500.0±0.0 | -500.0±0.0 |
| | BCQ-A | -500.0±0.0 | -500.0±0.0 | -500.0±0.0 | -500.0±0.0 | -500.0±0.0 |
| | PerSim-BCQ-5 | **-68.84**±15.2 | **-82.60**±7.00 | **-209.4**±206 | -498.7±1.89 | -500.0±0.0 |
| | CQL-P | -500.0±0.0 | -500.0±0.0 | -500.0±0.0 | -500.0±0.0 | -500.0±0.0 |
| | CQL-A | -500.0±0.0 | -500.0±0.0 | -500.0±0.0 | -500.0±0.0 | -500.0±0.0 |
| | PerSim-CQL-5 | **-44.70**±0.1 | **-49.80**±0.0 | **-63.20**±0.1 | -500.0±0.0 | -500.0±0.0 |

| Environment | Method | (2/0.5) | (10.0/0.5) | (18.0/0.5) | (10/0.85) | (10/0.15) |
|---|---|---|---|---|---|---|
| CartPole | BCQ-P | 44.80±34.0 | 58.21±58.0 | 56.92±56.0 | 57.91±53.0 | 36.40±36.0 |
| | BCQ-A | 43.90±16.4 | 18.70±13.1 | 7.200±0.84 | 21.10±5.81 | 39.50±12.1 |
| | PerSim-BCQ-5 | **80.04**±5.68 | **95.29**±29.5 | **63.86**±17.6 | **92.47**±19.9 | **57.82**±19.8 |
| | CQL-P | 39.90±29.9 | 72.30±41.7 | 67.80±44.4 | 77.90±26.2 | 148.7±17.1 |
| | CQL-A | 67.40±49.1 | 9.300±0.1 | 16.30±9.9 | 30.60±26.4 | 7.000±1.9 |
| | PerSim-CQL-5 | **81.80**±3.4 | **198.3**±1.3 | **200.0**±0.0 | **135.5**±11.1 | **190.0**±14.2 |

| Environment | Method | (0.3/1.7) | (1.7/0.3) | (0.3/0.3) | (1.7/1.7) | (1.0/1.0) |
|---|---|---|---|---|---|---|
| HalfCheetah | BCQ-P | -1.460±0.16 | -1.750±0.22 | -1.690±0.19 | -1.790±0.21 | -1.720±0.20 |
| | BCQ-A | -498.9±108 | -113.3±13.0 | -159.5±51.7 | -35.73±7.22 | -171.9±41.5 |
| | PerSim-BCQ-5 | **571.8**±40.3 | **22.19**±16.2 | -10.91±47.4 | **64.30**±2.60 | **157.0**±31.6 |
| | CQL-P | -481.5±25.9 | -442.2±56.4 | -672.4±17.6 | -254.5±39.6 | -418.2±23.4 |
| | CQL-A | -0.700±0.4 | -2.800±0.8 | -0.600±0.3 | -2.000±0.6 | -5.500±2.0 |
| | PerSim-CQL-5 | **1674**±135.2 | -38.50±35.9 | -402.2±113.0 | **76.70**±23.9 | **94.20**±102.8 |

## 5.5 Additional Experiments

**Generalizing to Unseen Agents (Appendix F.5).** We show how to extend our method to unseen test agents, i.e., agents for which we have no training trajectory. Our extension crucially relies on having learned a good agent specific latent factor representation. Through a case-study for the MountainCar environment, we verify that using this extension, PerSim can successfully simulate unseen agents.

**Robustness to Data Scarcity (Appendix F.6).** We investigate how the number of training agents affects PerSim's performance. We find that as we decrease training agents from $N = 250$ to $N = 25$, PerSim consistently achieves a higher reward than the other baselines across most agents. This ablation study directly addresses the robustness of PerSim to data scarcity, demonstrating that our principled framework is particularly suitable for the extreme data scarcity considered.

## 6 Conclusion

In this work, we investigate RL in an offline setting, where we observe a single trajectory across heterogeneous agents under an unknown, potentially sub-optimal policy. This is particularly challenging for existing approaches even in "solved" environments such as MountainCar and CartPole. PerSim offers a successful first attempt in simultaneously learning personalized policies across all agents under this data regime; we do so by first positing a principled low-rank latent factor representation, and then using it to build personalized simulators in a data-efficient manner.

**Limitations and Potential Impact.** Effectively leveraging offline datasets from heterogeneous sources (i.e., agents) for sequential decision-making problems will likely accelerate the adoption of RL. However, there is much to be improved in PerSim. For example, in environments like HalfCheetah where the transition dynamics of each agent are harder to learn, the performance of PerSim is not comparable with the online setting, where one gets to arbitrarily sample trajectories for each agent. Of course, our considered data regime is fundamentally harder. Therefore, understanding the extent to which we can improve performance, using our low-rank latent factor approach or a different methodology altogether, remains to be established. Additionally, a rigorous statistical analysis for this setting, which studies the effect of the degree of agent heterogeneity, the diversity of the samples collected, etc. remains important future work. We believe there are many fruitful inquiries under this challenging yet meaningful data regime for RL. Further, while PerSim is motivated by real-world problems, our empirical evaluation is limited to standard RL benchmarks where data collection and environment manipulation are feasible. Though PerSim's performance on standard RL benchmarks is encouraging, one cannot yet use PerSim as a out-of-the-box solution for critical real-world problems, without first designing a rigorous validation framework.

## Acknowledgements and Funding Disclosure

We would like to express our thanks to the authors of [29] Kimin Lee, Younggyo Seo, Seunghyun Lee, Honglak Lee, and Jinwoo Shin for for their insightful comments and feedback.

This work was supported in parts by the MIT-IBM project on time series anomaly detection, NSF TRIPODS Phase II grant towards Foundations of Data Science Institute, KACST project on Towards Foundations of Reinforcement Learning, and scholarship from KACST (for Abdullah Alomar).

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
