# Supplementary Materials

## A  Organization of Supplementary Materials

The supplementary materials consist of five main sections.

**Related Work.** In Appendix B, we give a detailed overview of the related literature.

**Proofs for Section 3.** In Appendix C, we give the proofs of Theorem 1 and Proposition 1.

**Algorithm and Implementation Details.** In Appendix D, we provide further details about the implementation and training procedure for PerSim and the RL methods we benchmark against.

**Detailed Experimental Setup.** In Appendix E, we detail the setup used to run our experiments. In Appendix E.1, we describe the OpenAI environments used. In Appendix E.2, we describe how we generate the offline training datasets for each environment.

**Additional Experimental Results.** In Appendix F, we provide more details for the experiments we run. Specifically, in Appendix F.1, we provide comprehensive results for the long-horizon prediction accuracy of model-based methods across all five test agents. In Appendix F.2, we provide comprehensive results for the achieved reward in the various environments for both the model-based and model-free methods across all five test agents. In Appendix F.3, we present additional visualizations of the latent agent factors. In Appendix F.4, we provide more details about the PerSim+BCQ/CQL experiments described in Section 5.4. In Appendix F.5, we describe and evaluate our proposed extension of PerSim to unseen test agents. In Appendix F.6, we evaluate PerSim's robustness to further data scarcity as we reduce the number of training agents.

## B  Related Work

**Model-based Online RL.** In model-based RL methods [50, 40, 19, 49, 21, 33, 12], the transition dynamics or simulator is learnt and subsequently utilized for policy learning. Compared to their model-free counterparts, model-based approaches, when successful, have proven to be far more data-efficient in terms of the number of samples required to learn a good policy and have shown to generalize better to unseen (state, action) tuples [10, 11, 26, 21, 17]. Recently, such methods have also been shown to effectively deal with agent heterogeneity, e.g., [29] learns a context vector using the recent trajectory of a given agent, with a common context encoder across all agents. Several recent works also utilize the meta-learning framework [13] to quickly adapt the model for model-based RL [39, 37, 36]. Thus far, the vast majority of the model-based RL literature has focused on the online setting, where transition dynamics are learned by adaptively sampling trajectories. Such online sampling helps these methods efficiently quantify and reduce uncertainty for unseen (state, action)-pairs. Further, there has been some work showing the success of online model-based RL approaches with offline data, with minimal change in the algorithm [19, 53]. This serves as additional motivation to compare with a state-of-the-art model-based RL method such as [29], which is designed to address agent heterogeneity.

**Model-free Offline RL.** As stated earlier, the offline RL paradigm [27, 31] is meant to allow one to leverage large pre-recorded (static) datasets to learn policies. Such methods are particularly pertinent for situations in which interacting with the environment can be costly and/or unethical, e.g., healthcare, autonomous driving, social/economic systems. The vast majority of offline RL methods are model-free [15, 24, 28, 32, 51, 4, 25]. Despite their rapidly increasing popularity, traditional offline RL methods suffer from "distribution shift", i.e., the policy learnt using such methods perform poorly on (state, action)-pairs that are unseen in the offline dataset [24, 15, 3, 31]. To overcome this challenge, offline RL methods design policies that are "close", in an appropriate sense, to the observed behavioural policy in the offline dataset [24, 51, 15]. They normally do so by directly regularizing the learnt policy (e.g. parameterized via the Q-function) based on the quantified level of uncertainty for a given (state, action)-pair. Most offline RL methods tend to be designed for the case where there is no agent heterogeneity. To study how much offline methods suffer if agent heterogeneity is introduced, we compare with one state-of-the-art offline RL method [15].

**Model-based Offline RL.** Model-based offline RL is a relatively nascent field. Two recent excellent works [23, 53] have shown that in certain settings, first building a model from offline data and then learning a policy outperforms state-of-the-art model-free offline RL methods on benchmark environments. By learning a model of the transition dynamics first, it allows such methods to trade-off the risk of leaving the behavioral distribution with the gains of exploring diverse states. However, the current inventory of model-based offline RL methods still require a large and diverse dataset

for each agent of interest—in fact, these methods restrict attention to the setting where there is just one agent of interest and one gets observations just from that one agent. Our approach effectively resolves the challenge via developing a principled and generic model representation. It is worth mentioning that several recent theoretical works have shown that structured MDPs (e.g., low-rank or linear transition model or value functions) can lead to provably efficient RL algorithms [52, 1, 20, 42], albeit in different settings. Extending the current model-based offline RL methods to work with sparse data from heterogeneous agents, possibly by building upon the latent low-rank tensor representation we propose, remains interesting future work.

## C    Theoretical Results

### C.1    Proof of Theorem 1

*Proof.* We will construct the function $h_d$ by partitioning the latent parameter spaces associated with agents, states, and actions. We then complete the proof by showing that $h_d$ is entry-wise close to $\tilde{f}_d$.

**Partitioning the latent spaces to construct $h_d$.** Fix some $\delta_1, \delta_3 > 0$. Since the latent row parameters $\theta_n$ come from a compact space $[0,1]^{d_1}$, we can construct a finite covering or partitioning $P_1(\delta_1) \subset [0,1]^{d_1}$ such that for any $\theta_n \in [0,1]^{d_1}$, there exists a $\theta_{n'} \in P_1(\delta_1)$ satisfying $\|\theta_n - \theta_{n'}\|_2 \leq \delta_1$. By the same argument, we can construct a partitioning $P_3(\delta_3) \subset [0,1]^{d_3}$ such that $\|\omega_a - \omega_{a'}\|_2 \leq \delta_3$ for any $\omega_a \in [0,1]^{d_3}$ and some $\omega_{a'} \in P_3(\delta_3)$.

For each $\theta_n$, let $p_1(\theta_n)$ denote the unique element in $P_1(\delta_1)$ that is closest to $\theta_n$. Similarly, define $p_3(\omega_a)$ as the corresponding element in $P_3(\delta_3)$ that is closest to $\omega_a$. We enumerate the elements of $P_1(\delta_1)$ as $\{\tilde{\theta}_1, ..., \tilde{\theta}_{|P_1(\delta_1)|}\}$. Analogously, we enumerate the elements of $P_3(\delta_3)$ as $\{\tilde{\omega}_1, ..., \tilde{\omega}_{|P_3(\delta_3)|}\}$. We now define $h_d$ as follows:

$$h_d(n,s,a) = \sum_{i=1}^{|P_1(\delta_1)|} \sum_{j=1}^{|P_3(\delta_3)|} \mathbb{1}(p_1(\theta_n) = \tilde{\theta}_i) \mathbb{1}(p_3(\omega_a) = \tilde{\omega}_j) f_d(\tilde{\theta}_i, \rho_s, \tilde{\omega}_j).$$

$\tilde{f}_d$ **is well approximated by** $h_d$**.** Here, we bound the maximum difference of any entry in $\tilde{f}_d$ from $h_d$. Using the Lipschitz property of $f_d$ (Assumption 1), we obtain for any $(n,s,a)$,

$$|\tilde{f}_d(n,s,a) - h_d(n,s,a)|$$
$$= \left| f_d(\theta_n, \rho_s, \omega_a) - \sum_{i=1}^{|P_1(\delta_1)|} \sum_{j=1}^{|P_1(\delta_3)|} \mathbb{1}(p_1(\theta_n) = \tilde{\theta}_i) \mathbb{1}(p_3(\omega_a) = \tilde{\omega}_j) f_d(\tilde{\theta}_i, \rho_s, \tilde{\omega}_j) \right|$$
$$= |f_d(\theta_n, \rho_s, \omega_a) - f_d(p_1(\theta_n), \rho_s, p_3(\omega_a))|$$
$$\leq L(\|\theta_n - p_1(\theta_n)\|_2 + \|\omega_a - p_3(\omega_a)\|_2)$$
$$\leq L(\delta_1 + \delta_3).$$

This proves that $\tilde{f}_d$ is entry-wise arbitrarily close to $h_d$.

**Concluding the proof.** It remains to write $h_d(n,s,a)$ as $\sum_{\ell=1}^{r} u_\ell(n) v_\ell(s,d) w_\ell(a)$ and bound the induced $r$. To that end, for $\ell = (i,j) \in [|P_1(\delta_1)|] \times [|P_3(\delta_3)|]$, we define

$$u_\ell(n) := \mathbb{1}(p_1(\theta_n) = \tilde{\theta}_i), \quad v_\ell(s,d) := f_d(\tilde{\theta}_i, \rho_s, \tilde{\omega}_j), \quad w_\ell(a) := \mathbb{1}(p_3(\omega_a) = \tilde{\omega}_j).$$

This allows us to write $r = |P_1(\delta_1)| \cdot |P_3(\delta_3)|$. Since each of the latent spaces is a unit cube of different dimensions, i.e., $[0,1]^x$ with $x \in \{d_1, d_3\}$, we can simply create partitions $P_1(\delta_1), P_3(\delta_3)$ by creating grid of cubes of size $\delta_1$ and $\delta_3$ respectively. In doing so, the number of such cubes will scale as $|P_1(\delta_1)| \leq C\delta_1^{-d_1}, |P_3(\delta_3)| \leq C\delta_3^{-d_3}$, where $C$ is an absolute constant. As such, $r \leq C\delta_1^{-d_1}\delta_3^{-d_3}$. Setting $\delta = \delta_1 = \delta_3$ completes the proof. □

### C.2    Proof of Proposition 1

*Proof.* To show that $r = 3$, it suffices to find $u(n), v(s_n, 1), v(s_n, 2), w(a_n) \in \mathbb{R}^3$ such that $h_d(n, s_n, a_n) = \sum_{\ell=1}^{r} u_\ell(n) v_\ell(s,d) w_\ell(a_n)$ for any $n \in [N]$, $s_n = [s_{n1}, s_{n2}] \in \mathcal{S}$, $a_n \in \mathcal{A}$, and $d \in \{1, 2\}$. In particular, we require that $u(n)$ can only depend on agent $n$, i.e., not on the action or state. Analogously, $v(s_n, 1)$ and $v(s_n, 2)$ can only depend on the state, and $w(a_n)$ can only depend on the

action. Towards this, consider the following factors:

$$u(n) = [1 \quad g_n \quad 1], \qquad w(a_n) = [1 \quad 1 \quad a_n],$$

$$v(s_n,1) = \left[ s_{n1} + s_{n2} \quad -\frac{\cos(3s_{n1})}{2} \quad \frac{1}{2} \right], \qquad v(s_n,2) = [s_{n2} \quad -\cos(3s_{n1}) \quad 1].$$

Recalling $h_1(n,s_n,a_n) = s_{n1} + s_{n2} - \frac{g_n\cos(3s_{n1})}{2} + \frac{a_n}{2}$ and $h_2(n,s_n,a_n) = s_{n2} - g_n\cos(3s_{n1}) + a_n$ completes the proof. □

# D   Algorithm and Implementation Details

## D.1   PerSim

**Step 1 Details: Learning Personalized Simulators.** As explained in Section 4, the personalized simulators are effectively trained by learning $g_u$, $g_v$, and $g_w$, which correspond to the agent, state, and action encoders, respectively. Below, we detail the architecture used for each function.

1. **Agent encoder: $g_u$.** We use a single layer that takes in a one-hot encoder of the agent and returns an $r$-dimensional latent factor.
2. **State encoder: $g_v$.** We use a multilayer perceptron (MLP) with 1 hidden layer of 256 ReLU activated nodes for both MountainCar and CartPole, and an MLP with 4 hidden layers each with 256 ReLU activated nodes for HalfCheetah.
3. **Action encoder: $g_w$.** In environments with discrete action spaces, i.e., MountainCar and CartPole, we use a single layer that takes in a one-hot encoder of the action and produce an $r$-dimensional latent factor. For HalfCheetah, we use an MLP with 2 hidden layers of 256 ReLU activated nodes.

We choose the tensor rank $r$ to be 3, 5, and 15 for the MountainCar, CartPole, and HalfCheetah environments, respectively. The choice is made via cross validation from the set $\{3,5,10,15,20,30\}$. Specifically, 20% of the data points (selected randomly from different trajectories) are set aside for validation in the hyper-parameter selection process. We train our simulators with a learning rate of 0.001, 300 epochs, and a batch size of 512 for HalfCheetah and MountainCar and 64 for CartPole. Please refer to the pseudo-code in Algorithm 1 for a detailed description of the training procedure.

---

**Algorithm 1** Training Personalized Dynamic Models

---

1: **Input:** Dataset $\mathcal{D}$, Rank $r$, Learning rate $\eta$, Batch size $B$, Number of epochs $K$
2: **Output:** $g_u(\cdot;\psi)$, $g_v(\cdot;\phi)$, $g_w(\cdot;\theta)$
3: Initialize $\psi$, $\phi$, and $\theta$
4: **for** each epoch **do** :
5:      **for** each batch **do** :
6:          **for** $i=1$ to B **do** :
7:              Sample $\{s_i,a_i,s_i',n_i\} \sim \mathcal{D}$
8:              Compute $\Delta s_i \leftarrow s_i' - s_i$
9:              Get agent latent factor $\widehat{u}(n_i) \leftarrow g_u(n_i;\psi)$
10:             Get state latent factor $\widehat{v}(s_i) := [\widehat{v}(s_i,d)]_{d \in [D]} \leftarrow g_v(s_i;\phi)$
11:             Get action latent factor $\widehat{w}(a_i) \leftarrow g_w(a_i;\theta)$
12:             Get the error estimate $\mathcal{L}_i \leftarrow \|\Delta s_i - \sum_{\ell=1}^{r} \widehat{u}_\ell(n_i) v_\ell(s_i) \widehat{w}_\ell(a_i)\|_2^2$
13:          **end for**
14:          Update $\psi \leftarrow \psi - \eta \nabla_\psi \frac{1}{B} \sum_{i=1}^{B} \mathcal{L}_i$
15:          Update $\phi \leftarrow \phi - \eta \nabla_\phi \frac{1}{B} \sum_{i=1}^{B} \mathcal{L}_i$
16:          Update $\theta \leftarrow \theta - \eta \nabla_\theta \frac{1}{B} \sum_{i=1}^{B} \mathcal{L}_i$
17:      **end for**
18: **end for**

---

**Step 2 Details: Learning a Decision-making Policy.** As outlined in Section 4, we use MPC to select the best action. Specifically, we sample $C$ candidate action sequences of length $h$, which we denote as $\{a_1^{(i)},...,a_h^{(i)}\}_{i=1}^{C}$. The actions are sampled using cross entropy in environments with continuous action spaces and random shooting in environments with discrete action space [9, 7].

Since the offline data may not span the entire state-action space, planning using a learned simulator without any regularization may result in "model exploitation" [31]. To overcome this issue, we gauge the model uncertainty, as is common in the literature, as follows. We train an ensemble of $M$ simulators

$\{g_u^{(m)}, g_s^{(m)}, g_a^{(m)}\}_{m=1}^M$. Then, for $i \in [C]$, we evaluate the average reward of performing the $i$-th action sequence, which we denote by $r^{(i)}$, using the estimates across the $M$ simulators. Specifically,

$$r^{(i)} = \frac{1}{M} \sum_{m=1}^M \sum_{t=1}^h R\left(\widehat{s}_t^{(m,i)}, a_t^{(i)}\right),$$

where $\widehat{s}_t^{(m,i)}$ is the predicted trajectory according to the $m$-th simulator and the sequence of actions $\{a_1^{(i)}, ..., a_h^{(i)}\}$, and $R$ is the reward function (which we assume is known, as is done in prior works [29, 23]). Finally, we choose the first element from the sequence of actions with the best *average* reward, i.e., the sequence $\{a_1^{(i^*)}, ..., a_h^{(i^*)}\}$, where $i^* = \mathrm{argmax}_{i \in [C]} r^{(i)}$.

For MountainCar and CartPole, we use random shooting to sample 1000 candidate actions with a planning horizon of 50. For HalfCheetah, we use the cross entropy method to sample 200 candidate actions with a planning horizon of 30. For all environments, we train $M = 5$ simulators.

### D.2 Benchmarking Algorithms

**Vanilla CaDM + PE-TS CaDM.** We use the implementation provided by the authors in [29].[2] To train on offline data, we modify the sampling procedure in the implementation. Specifically, we change it to sample from a replay buffer containing the recorded trajectories. Similar to our method, we use MPC with a planning horizon of 30 for HalfCheetah, and 50 for MountainCar and CartPole. We train the forward dynamic model, the backward dynamic model, and the context encoder for 20 iterations each with a maximum of 200 epochs and a learning rate of 0.001. For PE-TS, as is done in [29], we use an ensemble of five dynamics models, and use twenty particles for trajectory sampling.

**BCQ-P +BCQ-A.** We use the implementation provided by the authors in [15].[3] Specifically, we use discrete BCQ for MountainCar and CartPole, and continuous BCQ for HalfCheetah. For both BCQ-P and BCQ-A, we train the policy for $5.5 \times 10^5$ iterations.

**CQL-P +CQL-A.** We use the CQL implementation provided by the d3rlpy library [41].[4] Specifically, we use discrete CQL for MountainCar and CartPole, and continuous CQL for HalfCheetah. For discrete CQL, we set the number of critics to 3, the parameter $\alpha$ to 1, and we use a batch size of 512 and a learning rate of $10^{-4}$. For continuous CQL, we set the Lagrange threshold to 5, the policy learning rate to $3e-5$, and the critic learning rate to $3e-4$. We choose these parameters according to the guidelines the authors provide in [25].

**MOReL-P +MOReL-A.** We use the available open source implementation of MOReL [15].[5] We further extend the implementation to accommodate for environments with discrete actions. We use uncertainty penalty of -50 in MountainCar and CartPole and a penalty of -200 in HalfCheetah. For the dynamic model, we use an MLP with two hidden layers. Each layer has 128 neurons in MountainCar and CartPole, and 1024 neurons in HalfCheetah as done in the original paper.

## E Detailed Setup

### E.1 Environments

Table 4: Environment parameters used for experiments.

| Environment | Parameter range | Test agents | Policy training agents |
|---|---|---|---|
| MountainCar | gravity $\in [0.0001, 0.0035]$ | {0.0001, 0.0005, 0.0010, 0.0025, 0.0035} | {0.0003, 0.00075, 0.00175, 0.0025, 0.0030} |
| CartPole | length $\in [0.15, 0.85]$
force $\in [2.0, 18]$ | {(2.0,0.5), (10.0,0.5), (18.0, 0.5), (10.0,0.85), (10.0,0.15)} | {(6.0,0.5), (14.0,0.5), (10.0,0.5), (10.0,0.675), (10.0, 0.325)} |
| HalfCheetah | relative mass $\in [0.2,1.8]$
relative damping $\in [0.2,1.8]$ | {(0.3,1.7), (1.7,0.3), (0.3, 0.3), (1.7,1.7), (1.0,1.0)} | {(0.6,1.4), (1.4,0.6), (0.6, 0.6), (1.4,1.4)} |

[2]https://github.com/younggyoseo/CaDM

[3]https://github.com/sfujim/BCQ

[4]https://github.com/takuseno/d3rlpy

[5]https://github.com/SwapnilPande/MOReL

We perform experiments on three environments from the OpenAI Gym: two classical non-linear control environments, MountainCar and CartPole, and one Mujoco environment, HalfCheetah [47]. Next, we describe these three environments in detail.

**MountainCar.** In MountainCar, the goal is to drive a under-powered car to the top of a hill by taking the least number of steps.

- *Observation.* We observe $x(t), \dot{x}(t)$: the position and velocity of the car, respectively.
- *Actions.* There are three possible actions $\{0,1,2\}$: (0) accelerate to the left; (1) do nothing; (2) accelerate to the right.
- *Reward.* The reward is defined as
$$R(t) = \begin{cases} 1, & x(t) \geq 0.5 \\ -1, & \text{otherwise.} \end{cases}$$
- *Environment modification.* We vary the gravity within the range $[0.0001, 0.0035]$. Note that with a weaker gravity, the environment is trivially solved by directly moving to the right. On the other hand, with a stronger gravity, the car must drive left and right to build up enough momentum. See Table 4 for details about the parameter ranges and the test agents.

**CartPole.** In CartPole, a pole is attached to a cart moving on a frictionless track. The goal is to prevent the pole from falling over by moving the cart to the left or to the right, and to do so for as long as possible (maximum of 200 steps).

- *Observation.* We observe $x(t), \dot{x}(t), \theta(t), \dot{\theta}(t)$: the cart's position, its velocity, the pole's angle, and its angular velocity, respectively.
- *Actions.* There are two possible actions $\{0,1\}$: (0) push to the right; (1) push to the left.
- *Reward.* The reward is 1 for every step taken without termination. The environment terminates when the pole angle exceeds 12 degrees or when the cart position exceeds 2.4.
- *Environment modification.* As in [29], we vary the length of the pole and push force within the ranges $[0.15, 0.85]$ and $[2.0, 18.0]$, respectively. See Table 4 for details about the parameter ranges and the test agents.

**HalfCheetah.** In HalfCheetah, the goal is to move the cheetah as fast as possible. The cheetah's body consists of 7 links connected via 6 joints.

- *Observation.* We observe an 18-dimensional vector that includes the angle and angular velocity of all six joints, as well as the 3-D position and orientation of the torso. Additionally, as is done in previous studies [29, 23], we append the center of mass velocity to our state vector to enable computing the reward from observations.
- *Actions.* The action $a(t) \in [-1,1]^6$ represents the torque applied at the six joints.
- *Reward.* The reward is defined as
$$R(t) = v(t) - 0.05 \|a(t)\|^2,$$
where $v(t)$ is the center of mass velocity at time $t$.
- *Environment modification.* As in [29], we scale the mass of every link and the damping of every joint by factors $m$ and $d$, respectively. Specifically, we vary both $m$ and $d$ within the range $[0.2, 1.8]$. See Table 4 for details about the parameter ranges and the test agents.

### E.2 Offline Datasets

As stated in Section 5, we generate four offline datasets for each environment with varying "optimality" of the sampling policy. Specifically, we generate 500 trajectories (one per agent) for each environment as per the following sampling procedures:

**(i) Pure**. In the Pure procedure, actions are sampled according to a fixed policy (for each agent) that has been trained to achieve reasonably good performance. Specifically, for each environment, we first train a policy using online model-free algorithms for the training agents shown in Table 4. Specifically, we train these logging policies using DQN [34] for MountainCar and CartPole, and using TD3 for and HalfCheetah. We train these policies to achieve rewards of approximately -200, 120, 3000, for MountainCar, CartPole, and HalfCheetah respectively. Then, to sample a trajectory for each of the 500 agents, we use the policy trained on the training agent with the closest parameter value.

**(ii) Random**. Actions are selected uniformly at random.

**(iii) Pure-$\varepsilon$-20**. Actions are selected uniformly at random with probability of 0.2, and selected via the pure policy otherwise.

**(iv) Pure-$\varepsilon$-40**. Actions are selected uniformly at random with probability of 0.4, and selected via the pure policy otherwise.

See Table 5 for details about the reward observed for the five test agents using these four sampling procedures, and the average reward and trajectory length achieved across all 500 agents.

Table 5: Observed reward and trajectory length in the four sampled datasets in each environment. Agent 1 to Agent 5 refer to the five test agents. The average is taken across all 500 agents.

| Environment | Data | Observed Reward | | | | | | Average |
|---|---|---|---|---|---|---|---|---|
| | | Agent 1 | Agent 2 | Agent 3 | Agent 4 | Agent 5 | Average | Trajectory Length |
| MountainCar | Pure | -48.0 | -50.0 | -57.0 | -171.0 | -134.0 | -112.926 | 113.914 |
| | Random | -500.0 | -500.0 | -500.0 | -500.0 | -500.0 | -496.324 | 496.344 |
| | Pure-eps-2 | -46.0 | -54.0 | -73.0 | -165.0 | -140.0 | -155.252 | 156.214 |
| | Pure-eps-4 | -55.0 | -64.0 | -500.0 | -264.0 | -208.0 | -227.278 | 228.18 |
| CartPole | Pure | 197 | 200 | 193 | 179 | 200 | 185.14 | 186.14 |
| | Random | 59 | 23 | 10 | 26 | 17 | 22.39 | 23.39 |
| | Pure-eps-2 | 180 | 199 | 30 | 179 | 199 | 157.92 | 158.92 |
| | Pure-eps-4 | 38 | 23 | 11 | 170 | 14 | 92.80 | 93.80 |
| HalfCheetah | Pure | 522.40 | 1246.32 | 251.25 | 2646.85 | 1011.99 | 1894.10 | 1000.00 |
| | Random | -395.58 | -65.77 | -106.80 | -150.01 | -323.86 | -251.75 | 1000.00 |
| | Pure-eps-2 | 399.95 | 938.58 | 189.17 | 1742.85 | 1137.45 | 1121.52 | 1000.00 |
| | Pure-eps-4 | 508.11 | -115.94 | 128.64 | 1155.23 | 216.69 | 771.71 | 1000.00 |

# F Additional Experimental Results

## F.1 Detailed Prediction Error Results

In this section, we provide additional results for the prediction error experiments. As detailed in Section 5, we evaluate the accuracy of the learned transition dynamics for each of the five test agent, focusing on long-horizon model prediction. Specifically, we predict the next 50-step state trajectory given an initial state and an unseen sequence of 50 actions. The sequence of 50 actions are chosen according to an unseen test policy. Precisely , the test policies are fitted via DQN for MountainCar and CartPole, and via TD3 for and HalfCheetah, for the agent with the default covariate parameters. The test policies were trained to achieve an average rewards of -150, 150, 4000 for the MountainCar, CartPole, and HalfCheetah environments, respectively. As described in Section 5, we report the mean RMSE and the median $R^2$ across 200 trials. The results are summarized in Tables 6, 7, and 8 for MountainCar, CartPole, and HalfCheetah, respectively. Additionally, Figure 4 visualizes the prediction accuracy of PerSim up to 90-steps ahead predictions for two test agents in MountainCar and CartPole.

Table 6: Prediction error: MountainCar

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

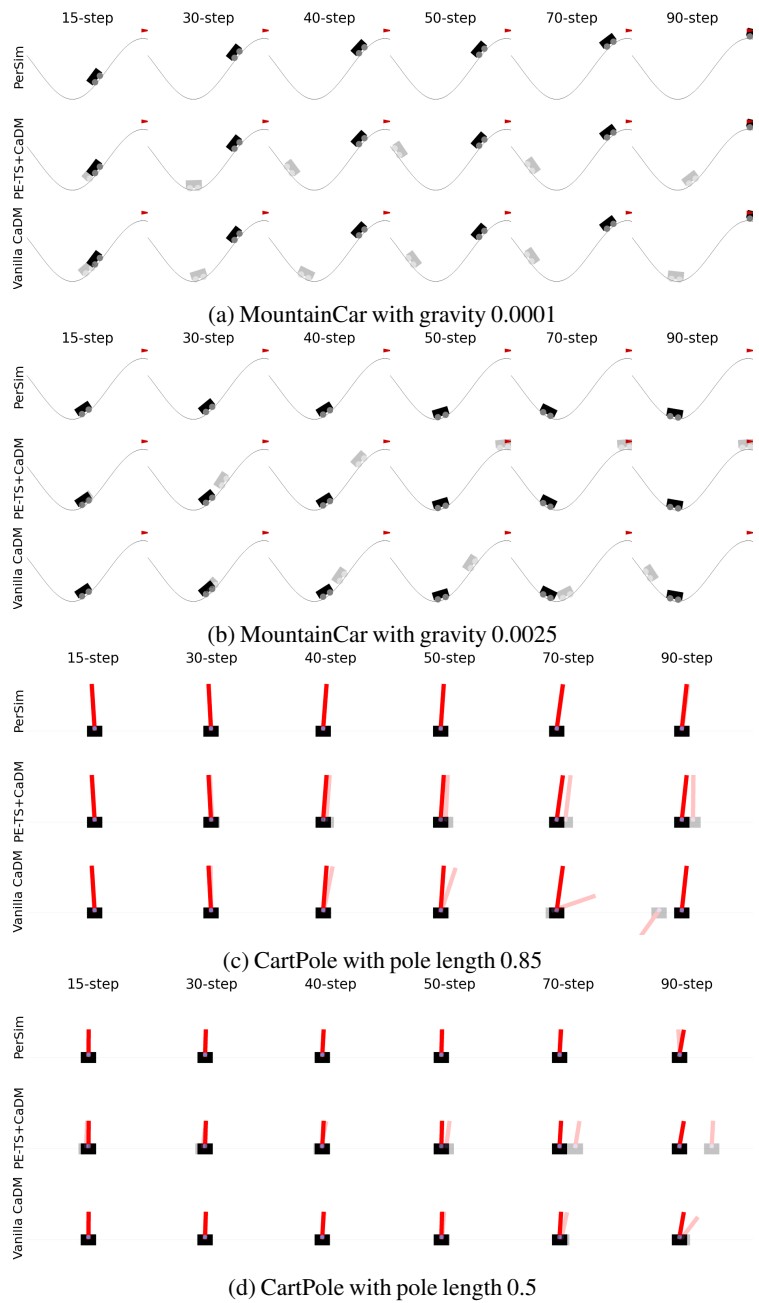

(a) MountainCar with gravity 0.0001

(b) MountainCar with gravity 0.0025

(c) CartPole with pole length 0.85

(d) CartPole with pole length 0.5

Figure 4: Visualization of the prediction accuracy of PerSim for two heterogeneous agents in MountainCar and CartPole, and how it compares with the two CaDM variants. Specifically, given an initial state and a sequence of actions, we predict future states for the next 90 steps. Ground-truth states and predicted states are denoted by the opaque and translucent objects, respectively.

## F.2   Detailed Average Reward Results

In this section, we show the full results for the experiments for the reward achieved in each environment. Specifically, we report the average reward achieved by PerSim and several state-of-the-art model-based and model-free offline RL methods on the three benchmark environments across 5 trials. We evaluate the performance of each method using the average reward over 20 episodes for the model-based methods and the average reward over 100 episodes for the model-free methods. We repeat each experiment five times and report the mean and standard deviation.

The results are summarized in Tables 9, 10, and 11 for MountainCar, CartPole, and HalfCheetah, respectively.

### Table 9: Average Reward: MountainCar

| Data | Method | Agent 1 0.0001 | Agent 2 0.0005 | Agent 3 0.001 | Agent 4 0.0025 | Agent 5 0.0035 |
|---|---|---|---|---|---|---|
| Pure | PerSim | $-56.80\pm1.83$ | $-74.30\pm6.59$ | $-114.1\pm16.1$ | $-189.4\pm6.44$ | $-210.6\pm4.27$ |
| | Vanilla CaDM | $-106.3\pm44.1$ | $-289.8\pm195$ | $-332.8\pm193$ | $-432.3\pm117$ | $-471.8\pm43.0$ |
| | PE-TS CaDM | $-74.23\pm16.5$ | $-119.1\pm44.4$ | $-361.3\pm240$ | $-492.3\pm13.3$ | $-500.0\pm0.0$ |
| | BCQ-P | $-67.60\pm22.3$ | $-68.60\pm19.6$ | $-79.20\pm14.7$ | $-267.8\pm202$ | $-295.1\pm180$ |
| | BCQ-A | $-44.79\pm0.08$ | $-50.50\pm0.40$ | $-63.52\pm0.19$ | $-380.7\pm170$ | $-500.0\pm0.0$ |
| | CQL-P | $-176.1\pm45.2$ | $-161.8\pm40.6$ | $-166.1\pm33.9$ | $-316.4\pm26.4$ | $-362.9\pm17.9$ |
| | CQL-A | $-44.60\pm0.0$ | $-49.70\pm0.0$ | $-63.30\pm0.3$ | $-500.0\pm0.0$ | $-499.3\pm0.7$ |
| | MOReL-P | $-46.00\pm1.1$ | $-53.20\pm3.3$ | $-220.3\pm162.6$ | $-500.0\pm0.0$ | $-500.0\pm0.0$ |
| | MOReL-A | $-373.0\pm33.5$ | $-488.8\pm4.9$ | $-499.4\pm0.2$ | $-500.0\pm0.0$ | $-500.0\pm0.0$ |
| Random | PerSim | $-57.70\pm5.63$ | $-74.30\pm6.39$ | $-120.4\pm2.17$ | $-186.6\pm4.25$ | $-210.1\pm4.48$ |
| | Vanilla CaDM | $-62.57\pm5.11$ | $-75.27\pm4.59$ | $-274.9\pm74.2$ | $-479.3\pm21.7$ | $-497.5\pm4.39$ |
| | PE-TS CaDM | $-82.00\pm3.47$ | $-115.7\pm7.63$ | $-472.1\pm48.3$ | $-500.0\pm0.0$ | $-500.0\pm0.0$ |
| | BCQ-P | $-500.0\pm0.0$ | $-500.0\pm0.0$ | $-500.0\pm0.0$ | $-500.0\pm0.0$ | $-500.0\pm0.0$ |
| | BCQ-A | $-500.0\pm0.0$ | $-500.0\pm0.0$ | $-500.0\pm0.0$ | $-500.0\pm0.0$ | $-500.0\pm0.0$ |
| | CQL-P | $-500.0\pm0.0$ | $-500.0\pm0.0$ | $-500.0\pm0.0$ | $-500.0\pm0.0$ | $-500.0\pm0.0$ |
| | CQL-A | $-500.0\pm0.0$ | $-500.0\pm0.0$ | $-500.0\pm0.0$ | $-500.0\pm0.0$ | $-500.0\pm0.0$ |
| | MOReL-P | $-45.00\pm0.3$ | $-50.40\pm0.5$ | $-68.20\pm5.4$ | $-500.0\pm0.0$ | $-500.0\pm0.0$ |
| | MOReL-A | $-495.0\pm3.9$ | $-480.6\pm3.1$ | $-500.0\pm0.0$ | $-500.0\pm0.0$ | $-500.0\pm0.0$ |
| Pure-$\varepsilon$-20 | PerSim | $-54.20\pm0.56$ | $-67.80\pm0.48$ | $-111.7\pm6.20$ | $-191.2\pm6.70$ | $-199.7\pm3.99$ |
| | Vanilla CaDM | $-56.73\pm4.20$ | $-70.70\pm1.54$ | $-148.7\pm11.6$ | $-463.2\pm57.5$ | $-478.9\pm35.8$ |
| | PE-TS CaDM | $-107.6\pm36.3$ | $-158.5\pm84.3$ | $-464.6\pm37.3$ | $-500.0\pm0.0$ | $-500.0\pm0.0$ |
| | BCQ-P | $-71.21\pm24.4$ | $-72.10\pm20.5$ | $-78.41\pm14.3$ | $-286.6\pm196$ | $-328.3\pm158$ |
| | BCQ-A | $-364.5\pm180$ | $-435.7\pm63.7$ | $-282.7\pm308$ | $-260.6\pm51.4$ | $-204.5\pm68.9$ |
| | CQL-P | $-79.40\pm16.1$ | $-78.80\pm13.9$ | $-86.50\pm10.8$ | $-357.9\pm12.5$ | $-407.6\pm15.3$ |
| | CQL-A | $-44.70\pm0.1$ | $-49.70\pm0.0$ | $-63.30\pm0.2$ | $-500.0\pm0.0$ | $-500.0\pm0.0$ |
| | MOReL-P | $-83.50\pm15.6$ | $-81.70\pm14.2$ | $-88.60\pm11.1$ | $-357.0\pm13.8$ | $-407.4\pm17.1$ |
| | MOReL-A | $-44.60\pm0.1$ | $-49.70\pm0.0$ | $-63.30\pm0.2$ | $-500.0\pm0.0$ | $-500.0\pm0.0$ |
| Pure-$\varepsilon$-40 | PerSim | $-54.60\pm0.55$ | $-71.10\pm1.89$ | $-115.7\pm4.80$ | $-189.7\pm7.14$ | $-200.3\pm2.26$ |
| | Vanilla CaDM | $-55.23\pm0.76$ | $-67.90\pm7.30$ | $-163.7\pm30.8$ | $-481.7\pm25.3$ | $-496.2\pm4.31$ |
| | PE-TS CaDM | $-102.3\pm20.3$ | $-120.7\pm18.8$ | $-476.0\pm41.5$ | $-500.0\pm0.0$ | $-500.0\pm0.0$ |
| | BCQ-P | $-50.01\pm7.50$ | $-57.10\pm10.3$ | $-66.11\pm3.91$ | $-373.6\pm180$ | $-352.0\pm211$ |
| | BCQ-A | $-94.87\pm0.88$ | $-80.03\pm38.5$ | $-329.6\pm242$ | $-358.7\pm201$ | $-486.5\pm20.6$ |
| | CQL-P | $-61.40\pm2.1$ | $-64.90\pm2.1$ | $-75.10\pm1.2$ | $-366.9\pm21.3$ | $-429.1\pm18.8$ |
| | CQL-A | $-44.70\pm0.0$ | $-49.70\pm0.0$ | $-63.30\pm0.4$ | $-500.0\pm0.0$ | $-490.1\pm9.6$ |
| | MOReL-P | $-61.30\pm2.3$ | $-65.80\pm1.2$ | $-75.00\pm1.4$ | $-373.2\pm19.3$ | $-428.5\pm21.0$ |
| | MOReL-A | $-44.70\pm0.0$ | $-49.70\pm0.0$ | $-63.30\pm0.4$ | $-500.0\pm0.0$ | $-492.1\pm9.3$ |
| | True env+MPC | $-53.95\pm4.10$ | $-72.43\pm7.80$ | $-110.8\pm23.8$ | $-182.9\pm22.9$ | $-197.5\pm20.7$ |

Table 10: Average Reward: CartPole

| Data | Method | Agent 1 (2/0.5) | Agent 2 (10.0/0.5) | Agent 3 (18.0/0.5) | Agent 4 (10/0.85) | Agent 5 (10/0.15) |
|------|--------|-----------------|--------------------|--------------------|--------------------|--------------------|
| Pure | PerSim | $199.7\pm_{0.58}$ | $198.7\pm_{0.86}$ | $198.5\pm_{1.16}$ | $193.8\pm_{4.28}$ | $192.0\pm_{2.28}$ |
| | Vanilla CaDM | $168.0\pm_{19.7}$ | $197.7\pm_{1.50}$ | $173.6\pm_{6.10}$ | $190.8\pm_{6.80}$ | $58.10\pm_{10.7}$ |
| | PE-TS CaDM | $92.30\pm_{44.8}$ | $200.0\pm_{0.0}$ | $200.0\pm_{0.0}$ | $193.6\pm_{8.30}$ | $127.5\pm_{9.50}$ |
| | BCQ-P | $166.2\pm_{39.3}$ | $187.4\pm_{14.7}$ | $187.2\pm_{15.0}$ | $181.2\pm_{13.5}$ | $182.8\pm_{15.0}$ |
| | BCQ-A | $65.40\pm_{67.5}$ | $138.0\pm_{80.3}$ | $79.20\pm_{79.9}$ | $79.20\pm_{69.5}$ | $132.1\pm_{85.0}$ |
| | CQL-P | $154.4\pm_{17.7}$ | $190.8\pm_{1.2}$ | $193.9\pm_{1.7}$ | $190.9\pm_{0.7}$ | $170.2\pm_{0.6}$ |
| | CQL-A | $122.8\pm_{63.4}$ | $189.4\pm_{6.6}$ | $199.3\pm_{0.7}$ | $179.4\pm_{20.6}$ | $193.6\pm_{5.4}$ |
| | MOReL-P | $35.80\pm_{1.4}$ | $90.40\pm_{26.2}$ | $94.20\pm_{25.1}$ | $96.60\pm_{24.1}$ | $66.40\pm_{24.5}$ |
| | MOReL-A | $33.70\pm_{3.8}$ | $16.00\pm_{6.5}$ | $16.20\pm_{0.6}$ | $27.50\pm_{0.1}$ | $10.10\pm_{0.7}$ |
| Random | PerSim | $197.7\pm_{7.82}$ | $189.5\pm_{4.28}$ | $190.3\pm_{4.74}$ | $193.0\pm_{6.60}$ | $185.7\pm_{3.49}$ |
| | Vanilla CaDM | $150.5\pm_{15.7}$ | $158.7\pm_{17.1}$ | $161.4\pm_{15.8}$ | $175.6\pm_{5.80}$ | $65.10\pm_{16.3}$ |
| | PE-TS CaDM | $88.60\pm_{18.5}$ | $194.0\pm_{2.00}$ | $197.6\pm_{2.20}$ | $196.1\pm_{3.00}$ | $171.0\pm_{21.7}$ |
| | BCQ-P | $44.80\pm_{34.0}$ | $58.21\pm_{58.0}$ | $56.92\pm_{56.0}$ | $57.91\pm_{53.0}$ | $36.40\pm_{36.0}$ |
| | BCQ-A | $43.90\pm_{16.4}$ | $18.70\pm_{13.1}$ | $7.200\pm_{0.84}$ | $21.10\pm_{5.81}$ | $39.50\pm_{12.1}$ |
| | CQL-P | $39.90\pm_{29.9}$ | $72.30\pm_{41.7}$ | $67.80\pm_{44.4}$ | $77.90\pm_{26.2}$ | $148.7\pm_{17.1}$ |
| | CQL-A | $67.40\pm_{49.1}$ | $9.300\pm_{0.1}$ | $16.30\pm_{9.9}$ | $30.60\pm_{26.4}$ | $7.000\pm_{1.9}$ |
| | MOReL-P | $50.20\pm_{7.9}$ | $59.10\pm_{17.8}$ | $57.80\pm_{17.8}$ | $68.90\pm_{17.4}$ | $40.40\pm_{12.4}$ |
| | MOReL-A | $35.80\pm_{0.4}$ | $20.70\pm_{1.0}$ | $14.30\pm_{1.4}$ | $27.50\pm_{0.7}$ | $10.60\pm_{0.2}$ |
| Pure-$\varepsilon$-20 | PerSim | $199.8\pm_{0.24}$ | $200.0\pm_{0.0}$ | $200.0\pm_{0.0}$ | $199.1\pm_{1.3}$ | $197.8\pm_{1.68}$ |
| | Vanilla CaDM | $171.1\pm_{38.1}$ | $195.3\pm_{3.00}$ | $180.7\pm_{4.30}$ | $193.4\pm_{2.10}$ | $64.20\pm_{10.0}$ |
| | PE-TS CaDM | $98.30\pm_{42.9}$ | $199.6\pm_{0.50}$ | $199.0\pm_{1.40}$ | $198.6\pm_{0.40}$ | $141.1\pm_{12.0}$ |
| | BCQ-P | $98.90\pm_{30.2}$ | $170.9\pm_{19.0}$ | $163.0\pm_{36.5}$ | $162.1\pm_{15.5}$ | $86.10\pm_{72.1}$ |
| | BCQ-A | $67.30\pm_{62.2}$ | $130.0\pm_{76.1}$ | $33.40\pm_{0.62}$ | $65.60\pm_{51.8}$ | $140.0\pm_{80.6}$ |
| | CQL-P | $163.7\pm_{13.6}$ | $197.4\pm_{2.9}$ | $198.6\pm_{2.4}$ | $198.1\pm_{2.6}$ | $190.4\pm_{6.6}$ |
| | CQL-A | $42.40\pm_{11.9}$ | $189.8\pm_{10.2}$ | $22.20\pm_{20.0}$ | $199.0\pm_{1.9}$ | $199.8\pm_{0.2}$ |
| | MOReL-P | $166.7\pm_{13.6}$ | $197.0\pm_{3.1}$ | $198.3\pm_{2.6}$ | $197.7\pm_{2.7}$ | $189.4\pm_{7.1}$ |
| | MOReL-A | $41.40\pm_{13.1}$ | $188.3\pm_{11.0}$ | $20.00\pm_{21.9}$ | $198.8\pm_{2.1}$ | $199.9\pm_{0.2}$ |
| Pure-$\varepsilon$-40 | PerSim | $199.9\pm_{0.18}$ | $199.8\pm_{0.20}$ | $199.3\pm_{1.34}$ | $198.0\pm_{1.21}$ | $197.4\pm_{1.72}$ |
| | Vanilla CaDM | $160.6\pm_{46.6}$ | $197.3\pm_{1.50}$ | $194.9\pm_{3.70}$ | $191.9\pm_{6.40}$ | $79.60\pm_{31.6}$ |
| | PE-TS CaDM | $91.90\pm_{67.6}$ | $199.8\pm_{0.20}$ | $200.0\pm_{0.0}$ | $197.0\pm_{1.40}$ | $143.5\pm_{17.5}$ |
| | BCQ-P | $28.90\pm_{6.80}$ | $24.97\pm_{12.8}$ | $27.90\pm_{25.9}$ | $31.80\pm_{25.9}$ | $18.50\pm_{11.1}$ |
| | BCQ-A | $34.60\pm_{1.55}$ | $23.20\pm_{17.8}$ | $7.180\pm_{0.76}$ | $47.71\pm_{48.7}$ | $23.20\pm_{9.44}$ |
| | CQL-P | $182.2\pm_{18.0}$ | $197.4\pm_{4.6}$ | $198.9\pm_{2.1}$ | $198.3\pm_{2.9}$ | $191.9\pm_{8.7}$ |
| | CQL-A | $20.70\pm_{0.7}$ | $25.70\pm_{14.3}$ | $15.30\pm_{7.3}$ | $134.2\pm_{10.2}$ | $9.700\pm_{8.1}$ |
| | MOReL-P | $178.9\pm_{18.7}$ | $196.8\pm_{5.0}$ | $198.7\pm_{2.3}$ | $197.9\pm_{3.1}$ | $190.6\pm_{9.3}$ |
| | MOReL-A | $20.70\pm_{0.8}$ | $23.90\pm_{15.5}$ | $16.60\pm_{7.6}$ | $135.4\pm_{11.2}$ | $10.80\pm_{8.8}$ |
| | True env+MPC | $200.0\pm_{0.0}$ | $200.0\pm_{0.0}$ | $200.0\pm_{0.0}$ | $198.4\pm_{7.20}$ | $200.0\pm_{0.0}$ |

Table 11: Average Reward: HalfCheetah

| Data | Method | Agent 1 (0.3/1.7) | Agent 2 (1.7/0.3) | Agent 3 (0.3/0.3) | Agent 4 (1.7/1.7) | Agent 5 (1.0/1.0) |
|---|---|---|---|---|---|---|
| Pure | PerSim | $1984 \pm_{763}$ | $997.0 \pm_{403}$ | $714.7 \pm_{314}$ | $113.5 \pm_{289}$ | $1459 \pm_{398}$ |
| | Vanilla CaDM | $50.31 \pm_{71.7}$ | $-134.0 \pm_{81.1}$ | $11.39 \pm_{171}$ | $-169.8 \pm_{67.5}$ | $331.3 \pm_{201}$ |
| | PE-TS CaDM | $481.1 \pm_{252}$ | $503.7 \pm_{181}$ | $553.0 \pm_{127}$ | $246.0 \pm_{261}$ | $840.1 \pm_{383}$ |
| | BCQ-P | $549.8 \pm_{322}$ | $2006 \pm_{153}$ | $-65.18 \pm_{92.8}$ | $2564 \pm_{70.2}$ | $2469 \pm_{67.2}$ |
| | BCQ-A | $-262.7 \pm_{96.6}$ | $-139.0 \pm_{236}$ | $165.6 \pm_{83.1}$ | $1649 \pm_{622}$ | $937.2 \pm_{221}$ |
| | CQL-P | $-353.5 \pm_{78.4}$ | $-453.6 \pm_{71.9}$ | $-476.7 \pm_{129}$ | $2037 \pm_{294}$ | $-145.1 \pm_{189}$ |
| | CQL-A | $-65.00 \pm_{105}$ | $-257.9 \pm_{35.9}$ | $-279.6 \pm_{34.4}$ | $689.3 \pm_{52.9}$ | $301.9 \pm_{98.9}$ |
| | MOReL-P | $-1297 \pm_{519}$ | $-1256 \pm_{627}$ | $-1470 \pm_{727}$ | $-1175 \pm_{592}$ | $-1256 \pm_{608}$ |
| | MOReL-A | $-726.2 \pm_{4.9}$ | $-666.1 \pm_{42.6}$ | $-841.7 \pm_{39.5}$ | $-599.6 \pm_{28.2}$ | $-688.6 \pm_{12.4}$ |
| Random | PerSim | $2124 \pm_{518}$ | $2060 \pm_{900}$ | $472.0 \pm_{56.9}$ | $565.2 \pm_{377}$ | $474.8 \pm_{344}$ |
| | Vanilla CaDM | $288.4 \pm_{32.4}$ | $362.9 \pm_{55.4}$ | $351.8 \pm_{34.9}$ | $358.4 \pm_{205}$ | $475.0 \pm_{102}$ |
| | PE-TS CaDM | $754.6 \pm_{242}$ | $744.5 \pm_{281}$ | $767.4 \pm_{214}$ | $555.4 \pm_{73.1}$ | $2486 \pm_{1488}$ |
| | BCQ-P | $-1.460 \pm_{0.16}$ | $-1.750 \pm_{0.22}$ | $-1.690 \pm_{0.19}$ | $-1.790 \pm_{0.21}$ | $-1.720 \pm_{0.20}$ |
| | BCQ-A | $-498.9 \pm_{108}$ | $-113.3 \pm_{13.0}$ | $-159.5 \pm_{51.7}$ | $-35.73 \pm_{7.22}$ | $-171.9 \pm_{41.5}$ |
| | CQL-P | $-481.5 \pm_{25.9}$ | $-442.2 \pm_{56.4}$ | $-672.4 \pm_{17.6}$ | $-254.5 \pm_{39.6}$ | $-418.2 \pm_{23.4}$ |
| | CQL-A | $-0.700 \pm_{0.4}$ | $-2.800 \pm_{0.8}$ | $-0.600 \pm_{0.3}$ | $-2.000 \pm_{0.6}$ | $-5.500 \pm_{2.0}$ |
| | MOReL-P | $-102.3 \pm_{45.7}$ | $-188.7 \pm_{37.1}$ | $-181.0 \pm_{67.7}$ | $-142.5 \pm_{26.8}$ | $-141.4 \pm_{7.3}$ |
| | MOReL-A | $-430.8 \pm_{195}$ | $-673.7 \pm_{39.3}$ | $-365.5 \pm_{97.4}$ | $-645.5 \pm_{53.0}$ | $-674.9 \pm_{28.9}$ |
| Pure-$\varepsilon$-20 | PerSim | $3186 \pm_{604}$ | $1032 \pm_{232}$ | $1120 \pm_{243}$ | $971.2 \pm_{916}$ | $1666 \pm_{930}$ |
| | Vanilla CaDM | $412.0 \pm_{152}$ | $31.92 \pm_{109}$ | $460.2 \pm_{159}$ | $60.33 \pm_{139}$ | $166.6 \pm_{71.8}$ |
| | PE-TS CaDM | $1082 \pm_{126}$ | $1125 \pm_{132}$ | $1067 \pm_{64.3}$ | $1098 \pm_{344}$ | $2843 \pm_{204}$ |
| | BCQ-P | $254.6 \pm_{352}$ | $406.7 \pm_{71.1}$ | $385.9 \pm_{57.1}$ | $-95.34 \pm_{65.8}$ | $738.0 \pm_{512}$ |
| | BCQ-A | $376.8 \pm_{102}$ | $84.66 \pm_{53.3}$ | $230.1 \pm_{10.0}$ | $1180 \pm_{87.3}$ | $617.5 \pm_{32.6}$ |
| | CQL-P | $838.7 \pm_{24.5}$ | $3155 \pm_{125}$ | $539.9 \pm_{313}$ | $1479 \pm_{51.0}$ | $3561 \pm_{170}$ |
| | CQL-A | $-15.50 \pm_{9.0}$ | $-73.00 \pm_{26.3}$ | $-108.0 \pm_{64.6}$ | $656.9 \pm_{181.6}$ | $357.6 \pm_{110}$ |
| | MOReL-P | $0.600 \pm_{210}$ | $-171.2 \pm_{125}$ | $-219.9 \pm_{76.7}$ | $-106.3 \pm_{18.6}$ | $-83.60 \pm_{110}$ |
| | MOReL-A | $-781.0 \pm_{37.9}$ | $-613.1 \pm_{49.9}$ | $-847.1 \pm_{64.7}$ | $-599.2 \pm_{4.6}$ | $-702.9 \pm_{26.8}$ |
| Pure-$\varepsilon$-40 | PerSim | $2590 \pm_{813}$ | $1016 \pm_{283}$ | $1365 \pm_{582}$ | $803.9 \pm_{912}$ | $724.9 \pm_{236}$ |
| | Vanilla CaDM | $465.6 \pm_{49.2}$ | $452.7 \pm_{130}$ | $720.0 \pm_{74.9}$ | $176.7 \pm_{359}$ | $952.8 \pm_{591}$ |
| | PE-TS CaDM | $1500 \pm_{246}$ | $1218 \pm_{221}$ | $1339 \pm_{54.8}$ | $1569 \pm_{306}$ | $3094 \pm_{825}$ |
| | BCQ-P | $78.25 \pm_{200}$ | $173.8 \pm_{189}$ | $417.1 \pm_{155}$ | $-56.12 \pm_{64.4}$ | $55.46 \pm_{128}$ |
| | BCQ-A | $269.2 \pm_{60.7}$ | $-181.5 \pm_{57.4}$ | $193.0 \pm_{31.8}$ | $636.4 \pm_{137}$ | $207.0 \pm_{106}$ |
| | CQL-P | $808.5 \pm_{240}$ | $1662 \pm_{220}$ | $-156.3 \pm_{119}$ | $1416 \pm_{71.8}$ | $1908 \pm_{461}$ |
| | CQL-A | $-6.200 \pm_{2.9}$ | $-386.0 \pm_{42.4}$ | $37.10 \pm_{154}$ | $1121 \pm_{95.6}$ | $1184 \pm_{604}$ |
| | MOReL-P | $8.500 \pm_{61.6}$ | $-114.2 \pm_{72.2}$ | $-195.9 \pm_{77.3}$ | $-66.60 \pm_{7.5}$ | $22.60 \pm_{28.7}$ |
| | MOReL-A | $-325.9 \pm_{17.1}$ | $-644.5 \pm_{18.8}$ | $-798.8 \pm_{130}$ | $-609.0 \pm_{16.8}$ | $-711.7 \pm_{19.1}$ |
| | True env+MPC | $7459 \pm_{171}$ | $42893 \pm_{6959}$ | $66675 \pm_{9364}$ | $1746 \pm_{624}$ | $36344 \pm_{7924}$ |

## F.3 Visualization of Agent Latent Factors

In this section, we visualize the learned agent latent factors associated with the 500 heterogeneous agents in each of the three benchmark environments. Specifically, we visualize the agent latent factors in MountainCar, as we vary the gravity (Figure 5a); CartPole, as we vary the pole's length and the push force (Figures 5c and 5b, respectively); HalfCheetah, as we vary the cheetah's mass and the joints' damping (Figures 5e and 5d, respectively).

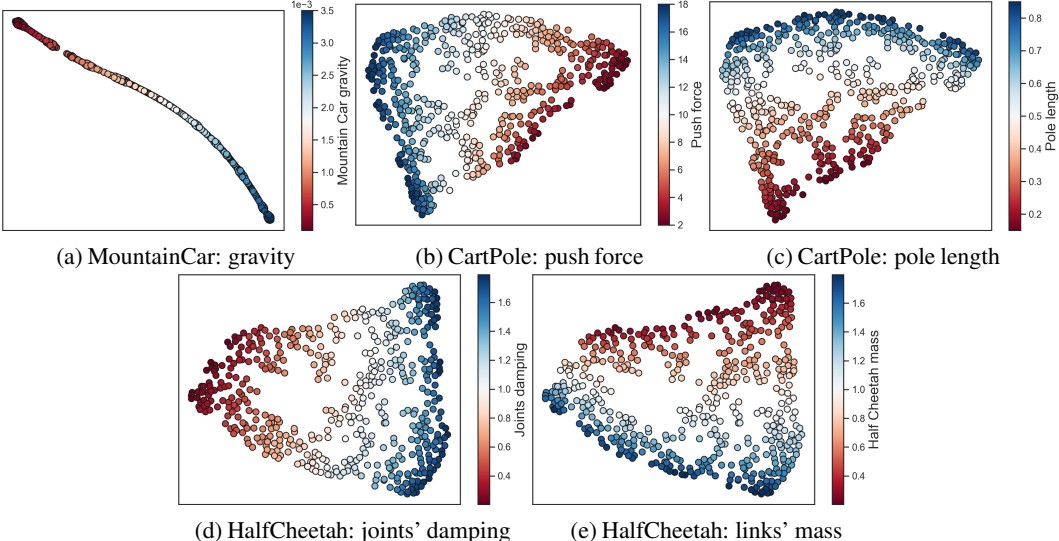

(a) MountainCar: gravity  (b) CartPole: push force  (c) CartPole: pole length

(d) HalfCheetah: joints' damping  (e) HalfCheetah: links' mass

Figure 5: t-SNE [48] visualization of the agent latent factors for the 500 heterogeneous agents in MountainCar, CartPole, and HalfCheetah. Colors indicate the value of the modified parameter in each environment (e.g., gravity in MountainCar). These figures demonstrate that the learned latent factors indeed capture the relevant information about the agents heterogeneity in all environments.

## F.4 Persim+BCQ/CQL: Experimental details

We evaluate PerSim's simulation efficacy by quantifying how much the simulated trajectories improve the performance of model-free RL methods such as BCQ and CQL. In particular, we use PerSim to generate synthetic trajectories for each agent of interest to augment the training data available for BCQ/CQL.

We carry out these experiments for the three environments: MountainCar, CartPole, and HalfCheetah. In all environments, as is done in previous experiments (see Section 5.1), we train PerSim using a single observed trajectory from each of the 500 training agents. For these trajectories, the actions are selected randomly, and the covariates of the training agents are selected as described in Section 5.1. Then, we use the trained simulators to produce 5 synthetic trajectories for each test agent. See Table 4 for information about the test agents in each environment and the range of covariates used for the training agents.

When generating these synthetic trajectories, we use MPC to choose the sequence of actions that maximizes the reward estimated by the simulator, as described in Appendix D.1. We use a horizon $h$ of 50 for MountainCar and CartPole, and a horizon $h$ of 30 for HalfCheetah. The only difference is that instead of choosing the first element from the sequence of actions with the best average reward, we choose the full sequence of actions, and repeat the sampling process until we have a full trajectory.

## F.5 Generalizing to Unseen Agents

**Setup.** In this experiment, we evaluate PerSim's ability to generalize to unseen agents. An advantage of the factorized approach of PerSim is that the heterogeneity of an agent is captured by its latent agent factors (see Table 5). Hence, the problem of generalizing to unseen agents boils down to accurately estimating the latent agent factors. To estimate these latent agent factors, we assume access to the covariates of the unseen agents, as well as a fraction $1 \geq p > 0$ of the covariates of the training agents.

With access to this information, we propose the following natural two-step procedure to estimate the latent agent factor: (1) Use a supervised learning method to learn a mapping between the (available) training agent covariates and the learned agent-specific latent factor in PerSim (see Figure 1); (2) Apply this mapping on the covariate data of an unseen test agent to estimate its latent agent factor, which is sufficient to build a personalized simulator for it.

We conduct this experiment for MountainCar, where we train PerSim using 500 agents, each with a gravity value selected uniformly at random from the range $[0.0001, 0.0035]$. We then evaluate PerSim's performance on 5 unseen agents selected as follow:

1. One unseen agent from the training range $[0.0001, 0.0035]$. Specifically, the one with gravity $0.002$.
2. Two agents outside the lower end of the range with gravity of $0.00008$ and $0.00005$.
3. Two agents outside the upper end of the range with gravity of $0.0037$, and $0.004$.

We first train PerSim on trajectories generated from the aforementioned 500 training agents, and carry out the experiments for trajectories generated via the random and pure policies. Then, we learn a mapping between the learned agent factors and covariates through an MLP with 2 hidden-layers each with 64 units. We assume access to a fraction $p \in \{1.0, 0.5, 0.25, 0.1\}$ of the covariates to train this function. We report PerSim's efficacy through the same two metrics we used before: prediction error and average reward for the five unseen agents.

**Results.** As Table 12 shows, in terms of prediction error, PerSim outperforms both Vanilla CaDM and PE-TS CaDM in both the random and pure datasets for all unseen test agents. Further, Table 13 shows that PerSim achieves the best reward among the three baselines (BCQ-P, Vanilla CaDM and PE-TS CaDM) for most unseen test agents. Pleasingly, these results are consistent as we vary $p \in \{1.0, 0.5, 0.25, 0.1\}$. That is, even with access to only 10% of the covariates of the training agents (i.e., $p = 0.1$), PerSim is able to simulate the unseen agents well. Note that PerSim is trained without access to any of the latent covariates (e.g., gravity in MountainCar), but to generalize to unseen agents, it requires access to some of the covariates to learn the mapping between these covariates and the latent agent factors. It is important to note that PerSim utilizes explicit knowledge of the covariates of the unseen test agents (and a subset of the training agents), which these other methods do not do in their current implementations. Indeed, it is our latent factor representation (in particular, the agent-specific latent factors) which seamlessly allows us to utilize these covariates to build simulators for the unseen agents.

Table 12: Prediction Error: MountainCar, Unseen Agents

| Data | Method | Agent 1 0.00005 | Agent 2 0.00008 | Agent 3 0.002 | Agent 4 0.0037 | Agent 5 0.004 |
|---|---|---|---|---|---|---|
| Random | PerSim(p=1.0) | 0.004 (1.00) | 0.004 (1.00) | 0.000 (1.00) | 0.001 (1.00) | 0.002 (1.00) |
| | PerSim(p=0.5) | 0.007 (0.99) | 0.006 (0.99) | 0.000 (1.00) | 0.001 (1.00) | 0.001 (1.00) |
| | PerSim(p=0.25) | 0.005 (0.99) | 0.004 (1.00) | 0.000 (1.00) | 0.001 (1.00) | 0.001 (1.00) |
| | PerSim(p=0.1) | 0.005 (0.99) | 0.005 (0.99) | 0.000 (1.00) | 0.000 (1.00) | 0.000 (1.00) |
| | Vanilla CaDM | 0.378 (0.13) | 0.373 (0.12) | 0.159 (0.13) | 0.329 (0.16) | 0.339 (0.15) |
| | PE-TS CaDM | 0.399 (0.06) | 0.351 (0.08) | 0.176 (0.07) | 0.193 (0.04) | 0.214 (0.05) |
| Pure | PerSim(p=1.0) | 0.192 (0.99) | 0.183 (0.98) | 0.029 (0.86) | 0.029 (0.90) | 0.034 (0.88) |
| | PerSim(p=0.5) | 0.039 (0.98) | 0.052 (0.99) | 0.029 (0.84) | 0.030 (0.89) | 0.036 (0.86) |
| | PerSim(p=0.25) | 0.061 (0.98) | 0.479 (0.98) | 0.029 (0.88) | 0.034 (0.87) | 0.039 (0.84) |
| | PerSim(p=0.1) | 0.043 (0.98) | 0.046 (0.98) | 0.033 (0.83) | 0.029 (0.90) | 0.034 (0.88) |
| | Vanilla CaDM | 0.213 (0.05) | 0.204 (0.04) | 0.199 (0.04) | 0.357 (0.03) | 0.362 (0.03) |
| | PE-TS CaDM | 0.432 (0.07) | 0.450 (0.09) | 0.187 (0.04) | 0.230 (0.06) | 0.232 (0.07) |

### F.6 Robustness to Data Scarcity

**Setup.** In this experiment, we address the robustness of PerSim to data scarcity. In particular, we decrease the number of observed agents from $N = 250$ to $N = 25$ in all three benchmarking environments. As is done in previous experiments, we compare with the two variants of CaDM and BCQ, and evaluate the performance on five test agents. We use trajectories generated by a random policy in MountainCar and HalfCheetah, and pure policy for CartPole. We use a pure policy for CartPole to ensure that each trajectory is not too short (see Table 5 for the average length of a trajectory in each environment under different policies). We perform these experiments five times, where in each time, the agent covariates are re-sampled from the covariates range. We report the average reward across the five trials and the corresponding standard deviations.

Table 13: Average Reward: MountainCar, Unseen Agents

| Data | Method | Agent 1 0.00005 | Agent 2 0.00008 | Agent 3 0.002 | Agent 4 0.0037 | Agent 5 0.004 |
|---|---|---|---|---|---|---|
| Random | PerSim(p=1.0) | -53.82±0.41 | -53.97±0.35 | -188.7±6.46 | -202.6±2.69 | -200.2±0.74 |
| | PerSim(p=0.5) | -53.98±0.93 | -54.57±0.39 | -192.3±3.61 | -207.8±2.53 | -204.0±1.15 |
| | PerSim(p=0.25) | -53.77±0.33 | -54.22±0.88 | -196.4±4.48 | -206.4±5.45 | -209.9±5.33 |
| | PerSim(p=0.1) | -53.25±0.56 | -54.23±1.05 | -198.2±1.38 | -207.3±0.86 | -208.6±3.40 |
| | Vanilla CaDM | -102.4±15.6 | -97.17±20.6 | -500.0±0.0 | -500.0±0.0 | -500.0±0.0 |
| | PE-TS CaDM | -82.60±20.2 | -96.07±5.10 | -500.0±0.0 | -500.0±0.0 | -500.0±0.0 |
| | BCQ-P | -500.0±0.0 | -500.0±0.0 | -500.0±0.0 | -500.0±0.0 | -500.0±0.0 |
| Pure | PerSim(p=1.0) | -75.17±12.8 | -74.73±11.4 | -179.3±4.74 | -203.7±4.13 | -201.4±5.72 |
| | PerSim(p=0.5) | -79.83±14.9 | -79.40±13.9 | -181.9±2.02 | -206.7±6.51 | -201.6±4.67 |
| | PerSim(p=0.25) | -77.28±13.4 | -77.73±19.7 | -178.4±2.77 | -199.7±3.41 | -209.1±4.83 |
| | PerSim(p=0.1) | -85.08±22.7 | -90.42±19.6 | -183.0±1.40 | -204.2±6.48 | -203.0±5.12 |
| | Vanilla CaDM | -52.63±1.05 | -54.90±2.60 | -446.8±51.2 | -494.1±9.30 | -494.1±10.3 |
| | PE-TS CaDM | -60.17±2.66 | -63.43±4.80 | -374.4±77.3 | -500.0±0.0 | -500.0±0.0 |
| | BCQ-P | -184.9±170 | -183.1±170 | -277.6±183 | -322.7±146 | -333.7±139 |

**Results.** We report the average reward achieved by PerSim and baselines in Tables 14, 15 and 16 for MountainCar, CartPole, and HalfCheetah, respectively. As demonstrated in the tables, even when we vary the number of trajectories, PerSim consistently achieves a higher reward than the other baselines across all agents in MountainCar and CartPole. In HalfCheetah, PerSim and PE-TS CaDM perform the best among the baselines. One thing to note is the high variance in HalfCheetah experiments, across all baselines, indicating the fundamental challenge faced when dealing with environments with both high-dimensional state space and limited data. Addressing such a challenge remains an interesting direction for future work.

Table 14: Average Reward: MountainCar with different number of training agents.

| Data | N | Method | Agent 1 0.0001 | Agent 2 0.0005 | Agent 3 0.001 | Agent 4 0.0025 | Agent 5 0.0035 |
|---|---|---|---|---|---|---|---|
| Random | 250 | PerSim | -53.70±0.41 | -66.50 ±1.21 | -116.6±3.18 | -192.3±1.23 | -199.6±3.40 |
| | | Vanilla CaDM | -59.90±1.61 | -78.13±7.11 | -332.6±41.3 | -467.8±16.2 | -500.0±0.0 |
| | | PE-TS CaDM | -73.66±3.15 | -106.8±7.15 | -473.8±37.0 | -500.0±0.0 | -500.0±0.0 |
| | | BCQ-P | -500.0±0.0 | -500.0±0.0 | -500.0±0.0 | -500.0±0.0 | -500.0±0.0 |
| Random | 100 | PerSim | -55.00±0.70 | -67.02±1.76 | -110.3±3.46 | -193.1±5.21 | -197.4±3.05 |
| | | Vanilla CaDM | -66.00±5.06 | -83.10±8.64 | -307.1±96.2 | -486.3±23.7 | -500.0±0.0 |
| | | PE-TS CaDM | -79.33±4.36 | -106.5±19.3 | -418.4±27.3 | -492.7±10.3 | -499.9±0.14 |
| | | BCQ-P | -500.0±0.0 | -500.0±0.0 | -500.0±0.0 | -500.0±0.0 | -500.0±0.0 |
| Random | 50 | PerSim | -54.20±0.90 | -66.40±0.17 | -110.2±8.65 | -188.7±5.25 | -199.6±3.23 |
| | | Vanilla CaDM | -58.80±1.40 | -66.37±0.35 | -131.8±17.6 | -497.2±4.85 | -500.0±0.0 |
| | | PE-TS CaDM | -67.86±7.15 | -79.73±6.37 | -290.5±76.9 | -458.4±26.8 | -498.5±2.12 |
| | | BCQ-P | -214.3±202 | -348.6±186 | -428.1±103 | -500.0±0.0 | -500.0±0.0 |
| Random | 25 | PerSim | -56.80±1.81 | -67.50±2.81 | -139.9±29.5 | -246.8±39.1 | -243.8±47.1 |
| | | Vanilla CaDM | -62.33±3.34 | -76.80±10.9 | -275.7±63.9 | -473.0±24.1 | -496.5±4.90 |
| | | PE-TS CaDM | -67.26±6.95 | -86.96±10.9 | -331.0±121 | -497.9±2.92 | -500.0±0.0 |
| | | BCQ-P | -500.0±0.0 | -500.0±0.0 | -500.0±0.0 | -500.0±0.0 | -500.0±0.0 |

Table 15: Average Reward: CartPole with different number of training agents.

| Data | N | Method | Agent 1 (2/0.5) | Agent 2 (10.0/0.5) | Agent 3 (18.0/0.5) | Agent 4 (10/0.85) | Agent 5 (10/0.15) |
|------|---|--------|---------|---------|---------|---------|---------|
| Pure | 250 | PerSim | 200.0±0.0 | 198.6±1.96 | 197.3±3.82 | 198.3±1.45 | 196.4±5.11 |
| | | Vanilla | 132.7±15.1 | 192.8±1.69 | 191.9±2.59 | 186.4±0.99 | 65.21±12.1 |
| | | PE-TS CaDM | 65.65±17.0 | 200.0±0.0 | 199.4±0.90 | 185.8±11.8 | 167.5±15.7 |
| | | BCQ-P | 130.2±1.31 | 169.6±21.6 | 173.3±7.94 | 167.3±8.81 | 179.6±2.71 |
| Pure | 100 | PerSim | 200.0±0.0 | 199.6±0.39 | 199.5±0.64 | 197.8±1.67 | 197.7±2.15 |
| | | Vanilla CaDM | 150.1±29.3 | 187.8±2.67 | 180.4±8.08 | 182.1±13.5 | 80.10±12.8 |
| | | PE-TS CaDM | 65.66±23.6 | 200.0±0.0 | 199.9±0.14 | 200.0±0.0 | 170.0±21.0 |
| | | BCQ-P | 90.67±52.1 | 49.18±32.0 | 119.0±78.6 | 108.9±70.1 | 81.30±66.6 |
| Pure | 50 | PerSim | 200.0±0.0 | 199.4±0.87 | 199.0±1.34 | 191.2±2.87 | 182.7±15.1 |
| | | Vanilla CaDM | 107.8±6.89 | 194.68±3.87 | 184.3±2.90 | 187.3±8.70 | 76.46±14.1 |
| | | PE-TS CaDM | 96.47±61.2 | 200.00±0.0 | 196.0±4.02 | 192.6±7.40 | 152.9±20.2 |
| | | BCQ-P | 121.0±30.0 | 70.96±15.8 | 122.4±18.6 | 145.6±29.0 | 100.1±44.2 |
| Pure | 25 | PerSim | 200.0±0.0 | 197.3±3.75 | 200.0±0.0 | 200.0±0.05 | 183.5±6.20 |
| | | Vanilla CaDM | 108.6±16.1 | 190.3±3.65 | 187.1±1.22 | 185.9±6.89 | 56.76±11.0 |
| | | PE-TS CaDM | 56.93±19.9 | 185.2±5.02 | 175.5±13.9 | 149.0±14.9 | 155.3±15.1 |
| | | BCQ-P | 135.9±4.07 | 55.27±47.4 | 154.7±9.99 | 152.5±16.7 | 144.9±25.3 |

Table 16: Average Reward: HalfCheetah with different number of training agents.

| Data | N | Method | Agent 1 (0.3/1.7) | Agent 2 (1.7/0.3) | Agent 3 (0.3/0.3) | Agent 4 (1.7/1.7) | Agent 5 (1.0/1.0) |
|------|---|--------|---------|---------|---------|---------|---------|
| Random | 250 | PerSim | 1688±1093 | 1415±1311 | 703.1±531 | 281.0±309 | 510.6±510 |
| | | Vanilla CaDM | 277.6±62.6 | 335.0±278 | 240.4±530 | 278.9±98.0 | 362.2±124 |
| | | PE-TS CaDM | 1006±556 | 1833±666 | 986.2±211 | 659.2±76.3 | 2303±571 |
| | | BCQ-P | -1.780±0.36 | -1.410±0.19 | -1.830±0.21 | -1.880±0.25 | -1.830±0.26 |
| Random | 100 | PerSim | 2072±284 | 1164±102 | 1115±493 | 903.3±398 | 1058±219 |
| | | Vanilla CaDM | 168.9±154 | 131.1±110 | 93.20±182 | 176.5±131 | 415.9±151 |
| | | PE-TS CaDM | 803.0±335 | 657.5±125 | 676.0±244 | 586.8±43.2 | 1484±934 |
| | | BCQ-P | -1.630±0.08 | -1.430±0.08 | -1.770±0.12 | -1.830±0.10 | -1.860±0.10 |
| Random | 50 | PerSim | 821.9±529 | 1984±58.2 | 103.0±122 | 66.49±194 | 701.2±112 |
| | | Vanilla CaDM | 236.0±234 | 670.0±167 | 68.55±220 | 248.7±44.4 | 802.8±355 |
| | | PE-TS CaDM | 496.4±166 | 1002±423 | 119.6±62.0 | 541.4±180 | 1895±989 |
| | | BCQ-P | -1.710±0.28 | -1.510±0.22 | -1.800±0.21 | -1.770±0.26 | -1.910±0.17 |
| Random | 25 | PerSim | 1110±328 | 1125±195 | 686.4±352 | 106.4±81.1 | 801.2±349 |
| | | Vanilla CaDM | 229.8±370 | 187.5±215 | -72.3±148 | 206.0±132 | 877.0±449 |
| | | PE-TS CaDM | 619.9±271 | 878.6±470 | 84.30±221 | 291.3±258 | 1464±913 |
| | | BCQ-P | -134.8±8.40 | -210.5±53.4 | -170.0±14.9 | -178.5±22.9 | -87.75±19.0 |