# OpenReview forum: "PerSim: Data-Efficient Offline Reinforcement Learning with Heterogeneous Agents via Personalized Simulators"
_NeurIPS.cc/2021/Conference — NeurIPS 2021 Poster_

### Official Review · Reviewer_6hLD · 2021-07-05

**Rating:** 6
**Confidence:** 4

**Summary:**

This paper proposes a new offline dataset setting: agents under different dynamics with rare sampling data (1 trajectory in the paper) for each; along with a latent factor decomposition method that motivated from the recommendation system domain. The method is interesting, the experiment of this paper is thorough and the writing is overall good.

**Limitations And Societal Impact:**

Limitations: the most limitation is the assumption of the method which needs further experiments on real-world problems. Also, the results on a high-dimensional environment are not that convincing.

Social Impact: The healthcare example is really attractive but when the latent personalized dynamic is well quantified, inappropriate usage of such models can lead to targeted attacking for specific people. However, these problems should be considered when it is appropriately designed for that application, not in this version.

**Main Review:**

Originality: The task is new as I know. The scenario and the motivating example are convincing and intriguing. The methodology is motivated by the usage of low-dimension latent factors decomposition in the recommendation domain and is interesting to be practiced in the RL scenario. The related works are mostly adequately cited.

Quality: The methodology is overall sound under the given assumption. The experimental results are thorough and seem good but are actually weak under the high-dimensional env (see detailed comments below) and may need more baseline methods. The work is complete, although it has some potential improvements.

Clarity: The writing is overall clear and well organized.

Significance: The problem seems to be new and is important to the application in practice (as the healthcare example that the authors proposed). However, this paper only works on a small range of simulated data. On these simulated environments, the proposed method gets good and robust results. The proposed thinking on the low-rank factor representation in RL is worth further considering.

Detailed comments:

Pro:

1. The problem is new and motivated; the proposed method solves the problem appropriately.
2. The idea for applying low-rank composition in RL is well put in an intuitive scenario.
3. The experiment and the analysis are thorough and the results are better than compared baselines.

Cons:

1. The assumption seems strong and may not be applicable in many real-world scenarios. In the paper the author explained their rationality for these assumptions by showing that it holds for a simple synthetic environment (MoutainCar). Maybe they can put it in a better context. Since all the experiments are conducted under simulated environments where the difference between the dynamics is the covariate, I am not much surprised by the result that they can share a low-rank representation. But, is it sensible for a real-world problem, e.g., the healthcare example?

2. The evaluation results on Halfcheetah is weak, although the proposed method beat all the baselines. Just seeing the last row of Table 2, where the oracle return of Halfcheetah takes a large gap with all compared methods and is less convincing. Also, the author should explain why the results on the last two dynamics (1.7/0.3, 0.3/0.3) are worse than the first (0.3/1.7) while all the environments contribute 1 trajectory to the dataset.

3. The comparison to baseline contains online MBRL and offline MFRL methods. So, why not offline MBRL methods are compared? Yes you mentioned that they are not designed not for this multi-task setting but BCQ is also not designed for it. Also, BCQ is not a good baseline since it has been outperformed by many other algorithms, such as CQL.

Considering the quality of the paper is overall good and the problem is new, I vote for a weak accept in this stage.

**Time Spent Reviewing:**

4

---

> ### Author Response · Authors · 2021-08-10
> **Response to Reviewer 6hLD**
>
>
> We thank the reviewer for their feedback and highlighting that the offline RL task we propose is novel, well-motivated, and of significance. We also thank the reviewer for pointing out that the approach we take of using a low-rank latent tensor decomposition, inspired from the recommendation systems literature, is interesting and intuitive for this setting. We now address the specific concerns the reviewer had.
>
> > The comparison to baseline contains online MBRL and offline MFRL methods. So, why not offline MBRL methods are compared? Yes you mentioned that they are not designed not for this multi-task setting but BCQ is also not designed for it. Also, BCQ is not a good baseline since it has been outperformed by many other algorithms, such as CQL.
>
> We thank the reviewer for this excellent suggestion. We agree and we will now include comparisons with both a state-of-the-art offline MBRL method, MOReL [1], and with Conservative Q-learning (CQL) [2] in our paper. As reference, we include a table in the main response (Table 1 and Table 2) where we add CQL and MOReL as baselines for the CartPole and MountainCar environments. We include both MOReL-P/CQL-P, and MOReL-A/CQL-A, in an analogous fashion to how BCQ-P and BCQ-A were trained. As can be seen, PerSim significantly outperforms MOReL-P, MOReL-A, CQL-A, CQL-P for both CartPole and MountainCar. This gives further evidence that MORel---despite being an offline MBRL method---and CQL are not able to effectively deal with the multi-task setting, i.e., heterogeneous agents. This supports the idea that the low-rank tensor decomposition used in PerSim is a good framework for this setting of heterogeneous agents and scarce data. We emphasize that we do significant hyper-parameter tuning for both MORel and CQL to get optimized performance for these two algorithms.
>
> Additionally, in Tables 3 and 4 in the main response, we see that when we use CQL-A on top of PerSim, it significantly improves the performance of CQL-A. This is in line with what we see with BCQ-A as well. Indeed, rather than consider CQL and BCQ as competing baselines, we view our work as complementary to these methods. Showing how one can improve the performance of offline RL methods such as CQL and BCQ, by training them with additional trajectories generated via PerSim, is a key contribution of this work. We believe one of the main reasons this leads to improved performance is that the simulators in PerSim carefully take agent heterogeneity into account.
>
> In our final manuscript, we will include a baseline with MOReL and CQL for the HalfCheetah and Ant environments as well. This will take a few weeks to run.
>
> > The evaluation results on Halfcheetah is weak, although the proposed method beat all the baselines. Just seeing the last row of Table 2, where the oracle return of Halfcheetah takes a large gap with all compared methods and is less convincing.
>
> We agree with the reviewer that for HalfCheetah, there is a large gap with the oracle setting where one has perfect access to the simulator. Indeed, a major point we hoped to make with this paper is that state-of-the-art: (i) offline RL methods (e.g. BCQ, CQL), (ii) model-based RL methods (e.g. CADM), and (iii) model-based offline RL methods (e.g. MOReL), all have a large gap when given access to the data setting we consider (one trajectory each of many heterogeneous agents). Indeed this gap is present for these previous methods even for MountainCar and CartPole, though it is not as pronounced as with HalfCheetah. We believe that empirically demonstrating that this gap exists for state-of-the-art RL methods to be a contribution of this paper, and we hope it spurs further research into closing this gap. It is worth noting that for most real-world settings, access to a perfect simulator is infeasible, and so this oracle setting is not a realistic benchmark.
>
> As pointed out by the reviewer, we re-emphasize that PerSim does outperform all of these state-of-the-art RL methods for this important data setting we consider, which should be considered a significant contribution in its own right. Further, that PerSim achieves results that are as good as the oracle setting for MountainCar and CartPole (while these other methods do not), indicates that the low-rank tensor decomposition architecture used in PerSim is likely a good way forward towards solving higher-dimensional environments like HalfCheetah.
>
> As further evidence, we include an additional experiment with the Ant environment, which is another high-dimensional OpenAI environment. See Table 5 in the main response for reference. As we can see, PerSim again outperforms the CaDM and BCQ baselines.
>
>
>
> > Since all the experiments are conducted under simulated environments where the difference between the dynamics is the covariate, I am not much surprised by the result that they can share a low-rank representation. But, is it sensible for a real-world problem, e.g., the healthcare example?
>
> We agree with the reviewer that when one attempts to use a method like PerSim for more realistic settings such as medicine, it is important to verify whether a low-rank latent representation exists. A data-driven way to do this would be to simply take medical trajectory data of patients and see what latent rank leads to the best prediction results. For example, in the MountainCar, CartPole, and HalfCheetah environments, the optimal latent rank was R = 3, 5 and 15, respectively. Further, there has been a recent line of work using low-rank matrix completion methods for medicine [3,4,5]. This provides further evidence that in many medical datasets, low-rank representations have been found to be useful/realistic.
>
> **Summary**
>
> We again thank the reviewer for their comments, and for constructive feedback that has allowed us to run additional useful baselines with MORel and CQL. We hope the reviewer takes these additional experiments into account in their revised scores. Thank you.
>
> **References:**
> 1. Kidambi, Rahul, et al. "Morel: Model-based offline reinforcement learning." arXiv preprint arXiv:2005.05951 (2020).
> 2. A. Kumar, A. Zhou, G. Tucker, and S. Levine. Conservative q-learning for offline reinforcement learning. arXiv preprint arXiv:2006.04779, 2020.
> 3. Chen, Yuxin, et al. "Inference and uncertainty quantification for noisy matrix completion." Proceedings of the National Academy of Sciences 116.46 (2019): 22931-22937.
> 4. Luo, Huimin, et al. "Computational drug repositioning using low-rank matrix approximation and randomized algorithms." Bioinformatics 34.11 (2018): 1904-1912.
> 5. Peng, Li, et al. "Improved low-rank matrix recovery method for predicting miRNA-disease association." Scientific reports 7.1 (2017): 1-10.

---

> > ### Comment · Reviewer_6hLD · 2021-08-11
> > **Unclarified questions**
> >
> > Thank the authors for the response.
> >
> > The response addresses some of my concerns, for example, the concern for comparing SoTA offline RL methods, and the concern for the low-rank representation in real-world data.
> >
> > However, it seems that you miss the question about "the author should explain why the results on the last two dynamics (1.7/0.3, 0.3/0.3) are worse than the first (0.3/1.7) while all the environments contribute 1 trajectory to the dataset." This is important to clarify your evaluation results.
> >
> > In addition, I am still not convinced by the high-dimensional results on Halfcheetah and Ant, since the evaluated returns on these tasks are so low that I think proper fine-tuning on baselines may provide competitive results. From my own experience on standard Mujoco experiments, Halfcheetah is much easier to get hundreds or even thousands of returns. Additional fine-tuning explanations on both your algorithm and the baselines may contribute more convincement to this.
> >
> > I really hope the authors provide more constructive feedback.
> >
> > Thanks.

---

> > > ### Author Response · Authors · 2021-08-13
> > > **Further Clarifications**
> > >
> > > We thank the reviewer for their feedback and prompt reply. Below, we address the reviewer's concerns.
> > >
> > > > In addition, I am still not convinced by the high-dimensional results on HalfCheetah and Ant, since the evaluated returns on these tasks are so low that I think proper fine-tuning on baselines may provide competitive results. From my own experience on standard Mujoco experiments, Halfcheetah is much easier to get hundreds or even thousands of returns. Additional fine-tuning explanations on both your algorithm and the baselines may contribute more convincement to this.
> > >
> > > We acknowledge that at first glance, the halfCheetah results look weak compared to the oracle setting. However, we believe that the seemingly weak performance is due to the challenging data availability we consider and **not** due to the lack of fine-tuning of the hyperparameters, which we did thoroughly for all baselines. Indeed, the performance of offline RL is highly dependent on the quality and quantity of the data available. The setting we are considering is uniquely challenging in two ways: (i) unlike many offline benchmarks in the literature, we consider data collected from heterogeneous agents/environments; (ii) from each environment/agent, we only sample one trajectory. To put this into perspective, in "halfcheetah-medium-expert", a dataset in D4RL (a popular offline RL benchmark), there are 2000 trajectories available from a single environment/agent.
> > >
> > > To illustrate the difficulty of the settings we consider and to verify that the problem is not related to the implementation used or lack of fine-tuning, we performed the following two experiments:
> > > In the first experiment, we train CQL, the offline RL algorithm recommended by the reviewer, on the D4RL dataset halfcheetah-medium-expert-v1. In this experiment, after proper fine-tuning of the parameters, we achieve an average reward of 8273 $\pm$ 1002 (across five different runs). Note that this is slightly higher than the results reported in the CQL and D4RL papers (7234 and 7466, respectively, see [1] [2]).
> > > In the second experiment, we apply the same method to both our Pure and Random datasets: once trained on the whole datasets (CQL-P) and once on only the trajectory of the same agent (CQL-A). Note that we tried different hyperparameters in these experiments as suggested by the authors (see Appendix F in [1]). Specifically, we varied the following parameters:
> > > - For the Q-function learning rate, we tried [1e-4, 3e-5], the two choices recommended by the authors.
> > > - For the policy learning rate, we tried [3e−5, 1e−4, 3e−4], the three choices recommended by the authors.
> > > - For the Lagrange threshold \tau, we tried [2,5,10]
> > > Below, we report the results of the best parameters (3e-5 for both Q-function and policy learning rate, and \tau = 5).
> > >
> > > | Data | Method | Agent 1 | Agent 2 | Agent 3 | Agent 4 | Agent 5 |
> > > |--------|--------|---------|---------|---------|---------|---------|
> > > | Pure | CQL-P | -428.8$\pm$49.2 |-444.1$\pm$82.9 |-546.8$\pm$98.8 |1801.7$\pm$00.9 |-183.6$\pm$127.2 |
> > > | Pure | CQL-A | -6.2$\pm$.2 |374.9$\pm$165.3 |-16.7$\pm$3.3 |-5.3$\pm$1.0 |-1.4$\pm$0.6|
> > > | Random | CQL-P | -485.2$\pm$21.4 |-481.1$\pm$40.9 |-759.8$\pm$134.7 |-240.5$\pm$63.7 |-429.5$\pm$5.7 |
> > > | Random | CQL-A |-0.6$\pm$0.4 |-0.4$\pm$0.4 |-1.1$\pm$0.7 |-1.0$\pm$0.1 |-0.7$\pm$0.1|
> > >
> > >
> > > As seen above, the reward achieved by CQL in our setting is much worse than its performance on the D4RL dataset. Since we use the same implementation of CQL in both settings, and given that we do extensive hyperparameter tuning as instructed by the authors of [1], the likely reason for these low scores is indeed the challenging data availability we consider.
> > >
> > > > However, it seems that you miss the question about "the author should explain why the results on the last two dynamics (1.7/0.3, 0.3/0.3) are worse than the first (0.3/1.7) while all the environments contribute 1 trajectory to the dataset." This is important to clarify your evaluation results.
> > >
> > > We thank the reviewer for giving us the chance to clarify our results, and we apologize for missing the question in our earlier response.
> > >
> > > While each environment contributes one trajectory, we believe that some environments are "harder" to simulate than others. In the halfCheetah example, we note that in these two particular environments, the damping of every joint is scaled by 0.3. With lower damping levels, the cheetah is more responsive to actions (torque applied). Hence, this leads to more variability in the trajectory of the cheetah for any given sequence of actions, compared to if the joint damping was 1.7, where the cheetah is far less responsive. We suspect this higher variability in the responsiveness of the 0.3 joint damping Cheetah makes it more challenging to approximate its dynamics with a low-rank tensor.
> > >
> > > This is indeed reflected in the prediction error results, where the prediction error of PerSim is relatively worse in these two agents. Despite this decrease, we emphasize the performance of PerSim on these two environments is still similar/better than both CaDM variants.
> > >
> > > Note a similar phenomenon is evident in Agent 5 in CarPole, where the performance of PerSim in terms of prediction error is worse relative to other agents. In Agent 5 in CartPole, the length of the pole is 0.15, which is less than the default length of 0.5. All remaining agents have a pole length of at least 0.5. Indeed, the results of PerSim on Agents 1-4 are close to perfect. However, for Agent 5, with a short pole length, the PerSim results are significantly worse. This is again because approximating the dynamics of a short pole is harder than a long pole. This is consistent with what happens to CaDM as well, which does relatively worse for Agent 5.
> > >
> > >  **Summary**
> > >
> > > We again thank the reviewer for both questions that helped us clarify our results. We will add this additional experiment above and an explanation of why 1.7/0.3 and 0.3/0.3 perform worse in our revised manuscript. We hope the reviewer takes these additional experiments into account in their revised scores. Thank you.
> > >
> > > **References**
> > >
> > > 1.  A. Kumar, A. Zhou, G. Tucker, and S. Levine. Conservative q-learning for offline reinforcement learning. arXiv preprint arXiv:2006.04779, 2020.
> > > 2. Fu, Justin, et al. "D4rl: Datasets for deep data-driven reinforcement learning." arXiv preprint arXiv:2004.07219 (2020).

---

> > > > ### Comment · Reviewer_6hLD · 2021-08-15
> > > > **Thanks for your work**
> > > >
> > > > Thanks for the detailed answers.
> > > >
> > > > The supplemented statement solved my concerns to some extent. It will be more convincing and clear to arrange these explanations and analyses into the paper.
> > > >
> > > > I am willing to keep my score under the current stage.

---

### Official Review · Reviewer_6vKu · 2021-07-11

**Rating:** 6
**Confidence:** 4

**Summary:**

The authors propose a novel model-based method for solving offline RL with heterogeneous agents. The authors show that under some continuity assumption, the dynamics model could be well approximated by a low-rank order-three functional tensor representation. In practice, it could be computed by the outputs of three neural networks, and one of which deals specifically with agent information. The proposed algorithm achieves robust and competitive performance on three control benchmarks and outperforms two state-of-the-art offline RL algorithms.

**Limitations And Societal Impact:**

The authors have provided a comprehensive list of limitations of their work and the proposed algorithm does not seem to have any negative societal impact.

**Main Review:**

Strength:

1. The idea of approximating the agent-specific dynamics model with the low-rank decomposition is novel. The authors provide theoretical justification for such representation, although it holds under some continuity assumption on the dynamics function. The additional example on MountainCar dynamics further solidifies the theory and provides good intuition on the algorithm design.

2. The experiment is comprehensive. The evaluation results show that the proposed algorithm achieves robust and competitive performance under various evaluation conditions and various data collecting methods, although the reviewer does have some further questions regarding the experiment setting (see below).

3. The paper overall is well organized. The presentation is clear, and the introduction of the method is detailed. It seems easy to reproduce the results and the authors also provide the source code.

Weakness/Minors:

1. Regarding the experiment setting: the current selected control tasks seem relatively easy compared to the main-stream benchmarks. Although the authors have already shown that even the state-of-the-art offline RL algorithms could fail on tasks as easy as MountainCar (under the proposed data collection scheme), it would be great to see how PerSim performs on more challenging tasks with more complicated dynamics, such as Hopper or Walker (HalfCheetah is considered relatively easy comparing with these two), or even tasks with high dimensional state space such as Ant.

2. The current presentation of the result table seems overwhelming at the first glance. It would be great if the authors could remind the readers about the evaluation contexts in the captions. Also, it would also be helpful if methods with statistically significantly better results could be highlighted.

3. The additional experiment on the scarcity of the data (e.g., 25 vs. 250 agents' trajectories) is reassuring. However, the reviewer is confused with the fact that only the random data collection scheme is included in the results. How is the performance different in the cases with the other data collection schemes?

4. It seems like that the training of PerSim relies on the assumption that the index of the agent is known (i.e., the input $n$ to the agent-specific network $g_u$). It may be a reasonable assumption under the current setting where each agent only has one trajectory but it could raise problems when the number of trajectories of each agent in the dataset is not fixed (then one needs to manually label the trajectories first, which seems to be extra information). Also, the reviewer may overlook but the reviewer does not follow how should the agent label be designed during test time?


**Time Spent Reviewing:**

6

---

> ### Author Response · Authors · 2021-08-10
> **Response to Reviewer 6vKu**
>
> We thank the reviewer for their positive feedback. In particular, for highlighting that modeling the heterogeneous agent dynamics via a low-rank tensor decomposition is novel with good theoretical backing, the experiments are comprehensive, and that PerSim achieves state-of-the-art results. We now address the specific concerns the reviewer had.
>
> > Regarding the experiment setting: the current selected control tasks seem relatively easy compared to the main-stream benchmarks. Although the authors have already shown that even the state-of-the-art offline RL algorithms could fail on tasks as easy as MountainCar (under the proposed data collection scheme), it would be great to see how PerSim performs on more challenging tasks with more complicated dynamics, such as Hopper or Walker (HalfCheetah is considered relatively easy comparing with these two), or even tasks with high dimensional state space such as Ant.
>
> We thank the reviewer for this constructive feedback. We now include an additional experiment with the Ant environment, which as stated by the reviewer is an environment with high-dimensional state. See Table 5 in the main response for reference. As we can see, PerSim again outperforms the CaDM and BCQ baselines. We focus our evaluation on where the various algorithms are trained on actions which are picked randomly for two reasons: (i) this better captures the setting where one must use sub-optimal offline data (i.e. data generated from sub-optimal policies) to learn an optimal policy; (ii) training an optimal policy to generate training data for PerSim, CaDM, and BCQ is time- and compute-intensive. We hope this initial experiment provides evidence that PerSim outperforms these baselines on environments with more complicated dynamics. We will include a comprehensive experiment with Ant in our final draft of the paper.
>
>
> As pointed out by the reviewer themselves, we would like to re-emphasize that a significant point we hoped to raise with this work was that even for simple environments, such as MountainCar and CartPole, state-of-the-art: (i) offline RL methods (e.g. BCQ, CQL [2]), (ii) model-based RL methods (e.g. CADM), and (iii) model-based offline RL methods (e.g. MOReL [1]) have poor performance when given scarce offline data of many heterogeneous agents. To add further evidence of how these previous methods suffer, in the main response (Tables 1 and 2), we include the results where we add CQL and MOReL as baselines  for the CartPole and MountainCar environments. CQL is another state-of-the-art offline RL method, and MOReL is a recent model-based offline RL method.  We include both MOReL-P/CQL-P, and MOReL-A/CQL-A, in an analogous fashion to how BCQ-P and BCQ-A were trained. As can be seen, PerSim significantly outperforms MOReL-P, MOReL-A, CQL-A, CQL-P for both CartPole and MountainCar. This gives further evidence that MORel---despite being an offline MBRL method---and CQL are not able to effectively deal with the multi-task setting, i.e., heterogeneous agents. We emphasize that we do significant hyper-parameter tuning for both MORel and CQL to get optimized performance for these two algorithms.
>
> Additionally, in Tables 3 and 4 in the main response,  we see that when we use CQL-A on top of PerSim, it significantly improves the performance of CQL-A. This is in line with what we see with BCQ-A as well. Indeed, rather than consider CQL and BCQ as competing baselines, we view our work as complementary to these methods. Showing how one can improve the performance of offline RL methods such as CQL and BCQ, by training them with additional trajectories generated via PerSim, is a key contribution of this work. We believe one of the main reasons this leads to improved performance is that the simulators in PerSim carefully take agent heterogeneity into account.
>
>
> > The current presentation of the result table seems overwhelming at the first glance. It would be great if the authors could remind the readers about the evaluation contexts in the captions. Also, it would also be helpful if methods with statistically significantly better results could be highlighted.
>
> Thank you for this great suggestion. In our revision, we will highlight statistically significant results and provide pointers/reminders to the evaluation contexts.
>
> > The additional experiment on the scarcity of the data (e.g., 25 vs. 250 agents' trajectories) is reassuring. However, the reviewer is confused with the fact that only the random data collection scheme is included in the results. How is the performance different in the cases with the other data collection schemes?
>
> We thank the reviewer for pointing out the usefulness of this additional experiment of exploring robustness to the number of trajectories we had. As stated earlier, we focus our evaluation on where the various algorithms are trained on actions which are picked randomly as this better captures the setting where one must use sub-optimal offline data (i.e. data generated from sub-optimal policies) to learn an optimal policy. In our revision, we will include the other data generating processes; however, given all experiments we have run so far, we do not expect the results to look very different than with random data generation.
>
> > It seems like that the training of PerSim relies on the assumption that the index of the agent is known (i.e., the input n to the agent-specific network gu). It may be a reasonable assumption under the current setting where each agent only has one trajectory but it could raise problems when the number of trajectories of each agent in the dataset is not fixed (then one needs to manually label the trajectories first, which seems to be extra information). Also, the reviewer may overlook but the reviewer does not follow how should the agent label be designed during test time?
>
> We agree that this algorithm does require providing a unique ID (UID) for each agent, which might require manual labeling. However in many real-world use cases such as medicine, e-commerce etc, having access to a UID for each agent is very realistic. Further, PerSim also works if we treat each trajectory as having come from a different agent, even if they do come from the same agent. That is, each trajectory can be given its own UID. The latent low-rank representation we propose will implicitly learn the similarity between the various trajectories via the agent latent factor associated with each trajectory. See Figure 5 in the paper for empirical evidence of this. Lastly, many previous methods that aim to learn across heterogenous agents require access to meaningful covariates about each agent. Clearly labeling UIDs for various trajectories is significantly less costly compared to having to collect meaningful covariates about each agent.
>
> **Summary**
>
> We again thank the reviewer for their comments, and for constructive feedback that has allowed us to run additional useful baselines with the Ant environment, and with MORel and CQL. And also for giving us the opportunity to clarify the relative difficulty of labeling the index of an agent. We hope the reviewer takes these additional experiments into account in their revised scores. Thank you.
>
> **References:**
> 1. Kidambi, Rahul, et al. "Morel: Model-based offline reinforcement learning." arXiv preprint arXiv:2005.05951 (2020).
> 2. A. Kumar, A. Zhou, G. Tucker, and S. Levine. Conservative q-learning for offline reinforcement learning. arXiv preprint arXiv:2006.04779, 2020.

---

> > ### Comment · Reviewer_6vKu · 2021-08-22
> > **Response to the authors**
> >
> > The reviewer appreciates the author's clarification. Although the reviewer would like to maintain their original evaluation towards an accept, they still would hope that the reviewer could highlight in the main text that the method indeed would require labeling of UID on the samples, which is extra information provided to the proposed method while comparing with the other offline baselines.

---

> > > ### Author Response · Authors · 2021-08-26
> > > **Response to reviewer**
> > >
> > > We thank the reviewer for their feedback.
> > >
> > > We will add a remark stating that PerSim requires access to a unique ID for each *trajectory*, and reference important real-world scenarios where access to such an ID is readily available. For example, (i) medicine where health trajectories are collected on a patient-by-patient basis; (ii) recommendation systems/e-commerce where trajectories are collected on a customer-by-customer basis. Also, we note that standard offline RL benchmarking datasets in D4RL have their data partitioned into distinct trajectories, i.e., they inherently organize their data on a trajectory-by-trajectory basis, so no additional manual labeling is required.
> > >
> > > Further, we note that while PerSim utilizes this readily available information, other baselines make use of additional information not used by PerSim. For example, to train its context encoder, both CaDM variants require access to continuous *sequential* (state → action → next state → next action → …..) tuples. The default parameter for how long they require this continuous trajectory to be is 10 contiguous time steps. In comparison, PerSim does not require continuous sequential trajectory data from a single environment, i.e., it can take in *out-of-order* (state → action → next state) tuples.
> > >
> > > We also note that many offline RL benchmarks have not been designed for the case where there are multiple agents, i.e., they assume all data is collected from a single agent, and so this question of trajectory ID does not show up. However, we explore how these baselines compare when given a single trajectory from an agent (corresponds to BCQ-A, CQL-A), or a collection of trajectories from many heterogeneous agents (corresponds to BCQ-P, CQL-P). The BCQ-A and CQL-A require knowing they are getting data from a single agent, which would require "manual" labeling of IDs for that specific agent's data; this is not required for BCQ-P and CQL-P. As we see in our experiments BCQ-A outperforms BCQ-P in a significant portion of experiments (~25-30% across different environments); the same holds for CQL-A and CQL-P.

---

### Official Review · Reviewer_p946 · 2021-07-16

**Rating:** 7
**Confidence:** 3

**Summary:**

The paper proposes PerSim, a model-based off-line reinforcement learning algorithm targeting a setting where each recorded trajectory is produced by acting in an environment with slightly different transition dynamics. PerSim first learns a simulator for each of those environments, taking advantage of a low-rank model similar to a recommender system to account for similarities between the environments. It then uses the simulator for a specific environment to perform model predictive control. The experiments compare against strong baselines, both model-free and model-based, and demonstrate gains both in terms of the average reward and in terms of simulator accuracy (in the latter case).

**Limitations And Societal Impact:**

yes

**Main Review:**

The paper addresses an important problem of using off-line reinforcement learning in a setting where data is scarce due to different demonstrations being recorded in different versions of the environment. While the experiments are performed on popular OpenAI gym environments, the authors provide a convincing motivating application in terms of personalized medical treatment planning. The architecture choice using latent factors with low-rank approximation makes a lot of sense. About half of the paper length is covering experimental evaluation, which I think is appropriate. The experiments are very comprehensive, demonstrating clear gains of PerSim against the baselines. I am not familiar with the CaDM and BQN baselines chosen for evaluation, but the selection seems appropriate in that it uses one model-free and one model-based baseline, with the former trained either on all trajectories in the dataset, or only the one recorded in the specific environment. Overall, this looks like a polished piece of work with a solid contribution.

**Time Spent Reviewing:**

5

---

> ### Author Response · Authors · 2021-08-10
> **Response to Reviewer p946**
>
>
> We thank the reviewer for such positive feedback.
>
> In line with the feedback we have received from other reviewers, we give a quick summary of the additional experiments we have run, which further emphasize the effectiveness of the approach used in PerSim.
>
> We now include comparisons with two additional baselines: with MOReL [1], which is a state-of-the-art offline model-based RL method; with Conservative Q-learning (CQL) [2], which is another state-of-the-art offline RL method. We include both MOReL-P/CQL-P, and MOReL-A/CQL-A, in an analogous fashion to how BCQ-P and BCQ-A were trained. As can be seen in Tables 1 and 2 in the main response,  PerSim significantly outperforms MOReL-P, MOReL-A, CQL-A, CQL-P for both CartPole and MountainCar. This gives further evidence that MORel---despite being an offline MBRL method---and CQL are not able to effectively deal with the multi-task setting, i.e., heterogeneous agents. This supports the idea that the low-rank tensor decomposition used in PerSim is a good framework for this heterogeneous agent and scarce data setting. We emphasize that we do significant hyper-parameter tuning for both MORel and CQL to get optimized performance for these two algorithms.
>
>
> Additionally, in the Tables 3 and 4 in the main response, we see that when we use CQL-A on top of PerSim, it significantly improves the performance of CQL-A. This is in line with what we see with BCQ-A as well. Indeed, rather than consider CQL and BCQ as competing baselines, we view our work as complementary to these methods. Showing how one can improve the performance of offline RL methods such as CQL and BCQ, by training them with additional trajectories generated via PerSim, is a key contribution of this work. We believe one of the main reasons this leads to improved performance is that the simulators in PerSim carefully take agent heterogeneity into account.
>
>
>
> We now include an additional experiment with the Ant environment, which as stated by the reviewer is an environment with high-dimensional state. See Table 5 in the main response  for reference. As we can see, PerSim again outperforms the CaDM and BCQ baselines. We focus our evaluation on where the various algorithms are trained on actions which are picked randomly as this better captures the setting where one must use sub-optimal offline data (i.e. data generated from sub-optimal policies) to learn an optimal policy. We hope this initial experiment provides evidence that PerSim outperforms these baselines on environments with more complicated dynamics. In our final manuscript, we will include a baseline with MOReL and CQL for HalfCheetah and Ant environments as well. This will take a few weeks to run.
>
> We again thank the reviewer for such positive comments. We hope the reviewer takes these additional experiments we run into account in their revised scores. Thank you.
>
> **References:**
> 1. Kidambi, Rahul, et al. "Morel: Model-based offline reinforcement learning." arXiv preprint arXiv:2005.05951 (2020).
> 2. A. Kumar, A. Zhou, G. Tucker, and S. Levine. Conservative q-learning for offline reinforcement learning. arXiv preprint arXiv:2006.04779, 2020.

---

### Official Review · Reviewer_Hdeq · 2021-07-18

**Rating:** 5
**Confidence:** 4

**Summary:**

This paper proposes a new model-based offline RL approach that tackles a particular offline RL problem where the dataset is generated by multiple agents and the data for each agent is scarce. To learn good policies from such heterogeneous dataset with limited per-agent data, the authors propose to learn per-agent personalized simulators (per-agent dynamics models) using latent low-rank factor representations with three encoders that encode the agent information, the state, and the action respectively. The authors show that the transition dynamics can be approximated with the low-rank order-three functional tensor representation with mild assumptions. Through experiments on gym tasks with heterogeneous agent data, the proposed method is shown to outperform prior online MBRL (adapted to the offline setting) and offline model-free RL method (BCQ).

**Limitations And Societal Impact:**

Yes

**Main Review:**

The paper studies a relatively new problem in offline RL, which is also quite practical since, in the real world, we would mostly have data collected by multiple agents. The proposed method is also novel as it takes the idea from recommendation systems and uses it to learn the dynamics model in the limited data setting. The authors also justify their approach through theoretical analysis, which makes the method theoretically grounded. The paper is also well written and easy to understand. I think the paper will be of reasonable significance to the offline RL community.

However, I do have a few concerns, which I will discuss as follows.

First, while the authors show that the method can approximate the true dynamics model to some extent, it is unclear if it will achieve better generalization guarantees and sample complexity compared to model learning trained with the maximum likelihood estimation objective. I think it would be important to show such a theoretical guarantee if the authors want to argue that the proposed algorithm can handle scarce data setting better than previous methods.

Moreover, I think the proposed problem setting is quite similar to the offline meta-RL setting, which also handles data collected by multiple agents with different dynamics and different reward functions. The authors should really discuss works in offline meta-RL and also compared the method to these works such as [1,2,3,4].

Finally, it seems that the empirical evaluations are not thorough. The authors only compare the method to an online MBRL method and BCQ. I think it would be important to compare the method to previous offline MBRL methods [5,6,7] and more recent model-free offline RL method [8,9,10]. To conduct fairer evaluations, I think the authors should also make previous offline RL methods conditioned on the task/agent. Furthermore, I'd like to see if the method can perform well on standard offline RL benchmarks such as D4RL, especially on domains with undirected multitask data such as antmaze and kitchen in D4RL.

Given the above comments, I would vote for a weak reject.

[1] Mitchell, E., Rafailov, R., Peng, X. B., Levine, S., & Finn, C. (2021, July). Offline Meta-Reinforcement Learning with Advantage Weighting. In International Conference on Machine Learning (pp. 7780-7791). PMLR.

[2] Dorfman, R., & Tamar, A. (2020). Offline meta reinforcement learning. arXiv e-prints, arXiv-2008.

[3] Li, L., Huang, Y., & Luo, D. (2021). Improved Context-Based Offline Meta-RL with Attention and Contrastive Learning. arXiv preprint arXiv:2102.10774.

[4] Li, L., Yang, R., & Luo, D. (2020). Efficient Fully-Offline Meta-Reinforcement Learning via Distance Metric Learning and Behavior Regularization. arXiv e-prints, arXiv-2010.

[5] R. Kidambi, A. Rajeswaran, P. Netrapalli, and T. Joachims. Morel: Model-based offline reinforcement learning. arXiv preprint arXiv:2005.05951, 2020.

[6] T. Yu, G. Thomas, L. Yu, S. Ermon, J. Zou, S. Levine, C. Finn, and T. Ma. Mopo: Model-based offline policy optimization. arXiv preprint arXiv:2005.13239, 2020.

[7] Argenson, A., & Dulac-Arnold, G. (2020). Model-based offline planning. arXiv preprint arXiv:2008.05556.

[8] A. Kumar, A. Zhou, G. Tucker, and S. Levine. Conservative q-learning for offline reinforcement learning. arXiv preprint arXiv:2006.04779, 2020.

[9] Sinha, S., & Garg, A. (2021). S4RL: Surprisingly Simple Self-Supervision for Offline Reinforcement Learning. arXiv preprint arXiv:2103.06326.

[10] Y. Wu, G. Tucker, and O. Nachum. Behavior regularized offline reinforcement learning. arXiv preprint arXiv:1911.11361, 2019


**Time Spent Reviewing:**

3 hours

---

> ### Author Response · Authors · 2021-08-10
> **Response to Reviewer Hdeq**
>
>
> We thank the reviewer for their constructive feedback, and for pointing out that the data setting we consider for offline RL is practical, the low-rank tensor decomposition idea is novel and theoretically grounded, and the paper will likely have reasonable significance to the offline RL community. We now address the specific concerns the reviewer had.
>
> > First, while the authors show that the method can approximate the true dynamics model to some extent, it is unclear if it will achieve better generalization guarantees and sample complexity compared to model learning trained with the maximum likelihood estimation objective. I think it would be important to show such a theoretical guarantee if the authors want to argue that the proposed algorithm can handle scarce data setting better than previous methods.
>
> We thank the reviewer for turning our attention towards maximum likelihood estimation (MLE) objectives for approximation the true dynamics model. To add context of how MLE fits into our framework, note that Theorem 1 in the paper establishes that the true underlying dynamics can be (approximately) represented as a  low-rank three order functional tensor. That is, the expected next state as a function of current state, action and unit decomposes as a   finite summation of a product of state, action and unit specific functions. The observed trajectories can be thought of as having access to a sparse sampling of such a functional tensor. Thus, the goal of learning the true dynamics effectively reduces to estimating this functional tensor accurately. Within this context of functional tensor completion, one can define the MLE estimator, which will need to take into account the observed sparsity pattern and "noise" in the observations.
>
> However, even in the the special case where rather than continuous state and action spaces, we have discrete space and actions, three-order tensor completion has been proven to be computationally hard for the MLE estimator [7]. Further, this computational hardness result exists for the "simpler" case where the entries of the tensor are observed independently and uniformly at random. However, in the setting we consider, there is "confounding", i.e., the observed entries of the tensor are generated by policies that pick actions (i.e., generate trajectories) in a very specific manner that is not at all well-approximated by a uniformly missing at random sparsity pattern. To the best of our knowledge, theoretical results for MLE for this setting we consider do not exist.
>
> Indeed, analyzing such "functional tensor completion" problems with "missing not at random data", as posed by this work is an important direction for future theoretical research in RL and more broadly. We shall include a discussion to this effect in our revised manuscript. Thank you for this terrific suggestion.
>
>
> > Moreover, I think the proposed problem setting is quite similar to the offline meta-RL setting, which also handles data collected by multiple agents with different dynamics and different reward functions. The authors should really discuss works in offline meta-RL and also compared the method to these works such as [1,2,3,4].
>
> We thank the reviewer for pointing us to these relevant references. We will add a discussion of these papers in our revision.
>
> However, we would like to note that these methods are very recent and hence it was infeasible to consider them at the time this manuscript was prepared for submission. Specifically, [1] and [4] are published very recently in ICML 2021 (July) and ICLR 2021 (May), respectively. [2,3] are still only available as ArXiv preprints. For this reason, understandably, we have no comparisons to these works included in the manuscript.
>
> Upon reading them over the past few days, as we understand, there are similarities to the setup of PerSim, but also some key differences.  In particular, these works attempt to learn across few tasks (i.e. a task is a heterogenous agent) with multiple trajectories per tasks. In contrast, we are intereated in learning across many tasks with a single trajectory per task.
>
>
> > Finally, it seems that the empirical evaluations are not thorough. The authors only compare the method to an online MBRL method and BCQ. I think it would be important to compare the method to previous offline MBRL methods [5,6,7] and more recent model-free offline RL method [8,9,10]. To conduct fairer evaluations, I think the authors should also make previous offline RL methods conditioned on the task/agent. Furthermore, I'd like to see if the method can perform well on standard offline RL benchmarks such as D4RL, especially on domains with undirected multitask data such as antmaze and kitchen in D4RL.
>
> We thank the reviewer for this excellent suggestion and for pointing us to relevant references. We agree and will now include comparisons with both a state-of-the-art offline model-based RL method, MOReL [5], and with Conservative Q-learning (CQL) [6] in our paper. As reference,  we include Tables 1 and 2 in the main response where we add CQL and MOReL as baselines for the CartPole and MountainCar environments. We include both MOReL-P/CQL-P, and MOReL-A/CQL-A, in an analogous fashion to how BCQ-P and BCQ-A were trained. Note that in BCQ-A,  CQL-A and MOReL-A, the offline method is trained using only data from the same agent (i.e. "conditioned on the task/agent") as the reviewer has suggested.
>
> As can be seen, PerSim outperforms MOReL-P, MOReL-A, CQL-A, CQL-P for both CartPole and MountainCar. This gives further evidence that MORel---despite being an offline MBRL method---and CQL are not able to effectively deal with the multi-task setting, i.e., heterogeneous agents. This supports the idea that the low-rank tensor decomposition used in PerSim is a good framework for this heterogeneous agent and scarce data setting. We emphasize that we perform hyper-parameter tuning for both MORel and CQL to get optimized performance for these two algorithms.
>
> Additionally, in Tables 3 and 4 in the main response, we  see that when we use CQL-A on top of PerSim, it significantly improves the performance of CQL-A. This is in line with what we see with BCQ-A as well. Indeed, rather than consider CQL and BCQ as competing baselines, we view our work as complementary to these methods. Showing how one can improve the performance of offline RL methods such as CQL and BCQ, by training them with additional trajectories generated via PerSim, is a key contribution of this work. We believe one of the main reasons this leads to improved performance is that the simulators in PerSim carefully take agent heterogeneity into account.
>
> We now include an additional experiment with the Ant environment, which is another OpenAI environment with high-dimensional state. See Table 5 in the main response for reference. As we can see, PerSim again outperforms the CaDM and BCQ baselines. We focus our evaluation on the data is generated by actions which are picked randomly as this better captures the setting where one must use sub-optimal offline data (i.e. data generated from sub-optimal policies) to learn an optimal policy. We hope this initial experiment provides evidence that PerSim outperforms these baselines on environments with more complicated  dynamics. In our final manuscript, we will include a baseline with MOReL and CQL for HalfCheetah and Ant environments as well. This will take a few weeks to run.
>
>
> We thank the reviewer for pointing to the D4RL dataset. One key limitation of the D4RL dataset, in particular antmaze and kitchen is that the notion of heterogeneity considered there is having the same agent, but the agent has to perform multiple tasks (e.g. ant has different start and end points in the maze, in kitchen different cooking related tasks). However, we consider the case where all agents have the same objective, but different transition dynamics. Hence, this dataset does not seem as relevant for the particular form of agent heterogeneity we consider.
>
> **Summary**
>
> We again thank the reviewer for their comments, and for constructive feedback that has allowed us to run additional useful baselines with MORel and CQL, and the Ant environment. We hope the reviewer takes these additional experiments into account in their revised scores. Thank you.
>
> **References:**
>
> 1. Mitchell, E., Rafailov, R., Peng, X. B., Levine, S., & Finn, C. (2021, July). Offline Meta-Reinforcement Learning with Advantage Weighting. In International Conference on Machine Learning (pp. 7780-7791). PMLR.
> 2. Dorfman, R., & Tamar, A. (2020). Offline meta reinforcement learning. arXiv e-prints, arXiv-2008.
> 3. Li, L., Huang, Y., & Luo, D. (2021). Improved Context-Based Offline Meta-RL with Attention and Contrastive Learning. arXiv preprint arXiv:2102.10774.
> 4. Li, L., Yang, R., & Luo, D. (2020). Efficient Fully-Offline Meta-Reinforcement Learning via Distance Metric Learning and Behavior Regularization. arXiv e-prints, arXiv-2010.
> 5. Kidambi, Rahul, et al. "Morel: Model-based offline reinforcement learning." arXiv preprint arXiv:2005.05951 (2020).
> 6. A. Kumar, A. Zhou, G. Tucker, and S. Levine. Conservative q-learning for offline reinforcement learning. arXiv preprint arXiv:2006.04779, 2020.
> 7. Barak, Boaz, and Ankur Moitra. "Noisy tensor completion via the sum-of-squares hierarchy." Conference on Learning Theory. PMLR, 2016.

---

> > ### Author Response · Authors · 2021-08-19
> > **Feedback**
> >
> > Dear Reviewer Hdeq:
> >
> > Thank you for your feedback and comments. Please let us know if you have any further comment after you look at the above response. We appreciate your help in improving our manuscript.
> >
> > Best.
> > Authors.

---

### Author Response · Authors · 2021-08-10
**Response: Results for Additional Experiments.**

# Additional Experimental Results

We thank the reviewers for their constructive feedback. In addition to the responses to individual reviewers below, we provide a summary of all the additional experiments we have run, which further emphasize the effectiveness of the approach used in PerSim. Due to space constraints, we present these results in the main response, and discuss them in the individual responses in the appropriate place.

## MOReL and CQL as baselines
We now include comparisons with two additional baselines : with MOReL [1], which is a state-of-the-art offline model-based RL method; with Conservative Q-learning (CQL) [2], which is another state-of-the-art offline RL method. We include both MOReL-P/CQL-P, and MOReL-A/CQL-A, in an analogous fashion to how BCQ-P and BCQ-A were trained. Below, we present the results for both CartPole and MountainCar.

### Table 1: MOReL and CQL results on CartPole
| Data   | Method       | Agent 1                       | Agent 2                       | Agent 3                       | Agent 4                       | Agent 5                       |
|--------|--------------|-------------------------------|-------------------------------|-------------------------------|-------------------------------|-------------------------------|
| Pure   | PerSim       | **199.7**$\pm$0.58 | **198.7**$\pm$0.86   | **198.5**$\pm$1.16   | **193.8**$\pm$4.28 | **192.0**$\pm$2.28 |
|        | CQL-P        | 154.4$\pm$17.7   | 190.8$\pm$1.2    | 193.9$\pm$1.7    | 190.9$\pm$0.7    | 170.2$\pm$0.6    |
|        | CQL-A        | 122.8$\pm$63.4   | 189.4$\pm$6.6    | **199.3**$\pm$0.9    | 179.4$\pm$20.6   | **193.6**$\pm$5.4    |
|        | MOReL-P      | 35.8$\pm$1.4     | 90.4$\pm$26.2    | 94.2$\pm$25.1    | 96.6$\pm$24.1    | 66.4$\pm$24.5    |
|        | MOReL-A      | 33.7$\pm$3.8     | 16.0$\pm$6.5     | 16.2$\pm$0.6     | 27.5$\pm$0.1     | 10.1$\pm$0.7     |
| Random | PerSim       | **197.7**$\pm$7.82 | **189.5**$\pm$4.28   | **190.3**$\pm$4.74   | **193.0**$\pm$6.60   | **185.7**$\pm$3.49 |
|        | CQL-P        | 39.9$\pm$29.9    | 72.3$\pm$41.7    | 67.8$\pm$ 44.4   | 77.9$\pm$26.2    | 148.7$\pm$ 17.1  |
|        | CQL-A        | 67.4$\pm$49.1    | 9.3$\pm$0.1      | 16.3$\pm$9.9     | 30.6$\pm$26.4    | 7.0 $\pm$1.9     |
|        | MOReL-P      | 50.2$\pm$7.9     | 59.1$\pm$17.8    | 57.8$\pm$17.8    | 68.9$\pm$17.4    | 40.4$\pm$12.4    |
|        | MOReL-A      | 35.8$\pm$0.4     | 20.7$\pm$1.0     | 14.3$\pm$1.4     | 27.5$\pm$0.7     | 10.6$\pm$0.2     |



 ### Table 2: MOReL and CQL results on CartPole

| Data            | Method          | Agent 1                          | Agent 2                          | Agent 3                          | Agent 4                          | Agent 5                          |
| --------        | --------------- | -------------------------------- | -------------------------------- | -------------------------------- | -------------------------------- | -------------------------------- |
| Pure            | PerSim          | -56.80$\pm$1.83                  | -74.30$\pm$6.59                  | -114.1$\pm$.1                    | **-189.4**$\pm$6.44              | **-210.6**$\pm$4.27              |
|                 | CQL-P           | -176.1$\pm$45.2                  | -161.8$\pm$40.6                  | -166.1$\pm$33.9                  | -316.4$\pm$26.4                  | -362.9$\pm$17.9                  |
|                 | CQL-A           | **-44.6**$\pm$0.0                | **-49.7**$\pm$0.0                | **-63.3**$\pm$0.3                | -500.0$\pm$0.0                   | -499.3$\pm$0.7                   |
|                 | MOReL-P         | -46.0$\pm$1.1                    | -53.2$\pm$3.3                    | -220.3$\pm$2.6                   | -500.0$\pm$0.0                   | -500.0$\pm$0.0                   |
|                 | MOReL-A        | -373.0$\pm$33.5 |-488.8$\pm$4.9 |-499.4$\pm$0.2 |-500.0$\pm$0.0 |-500.0$\pm$0.0 |
| Random          | PerSim          | **-57.70**$\pm$5.63                  | -74.30$\pm$6.39                  | **-120.4**$\pm$2.17                  | **-186.6**$\pm$4.25              | **-210.1**$\pm$4.48              |
|                 | CQL-P           | -500.0$\pm$ 0.0                  | -500.0$\pm$ 0.0                  | -500.0$\pm$ 0.0                  | -500.0$\pm$ 0.0                  | -500.0$\pm$ 0.0                  |
|                 | CQL-A           | -500.0$\pm$ 0.0                  | -500.0$\pm$ 0.0                  | -500.0$\pm$ 0.0                  | -500.0$\pm$ 0.0                  | -500.0$\pm$ 0.0                  |
|                 | MOReL-P         | **-56.0**$\pm$7.3                | **-50.4**$\pm$0.5                | -168.2$\pm$5.4                | -500.0$\pm$0.0                   | -500.0$\pm$0.0                   |
|                 | MOReL-A        | -495.0$\pm$3.9 |-480.6$\pm$3.1 |-500.0$\pm$0.0 |-500.0$\pm$0.0 |-500.0$\pm$0.0 |


## CQL + PerSim
Additionally, in the table below see that when we use CQL-A on top of PerSim, it significantly improves the performance of CQL-A. This is in line with what we see with BCQ-A as well. Below, we present the results for these experiments for both MountainCar and CartPole.



### Table 3: CQL + PerSim: MountainCar (Random data)
| | Method     | 0.0001                     | 0.0005                     | 0.001                      | 0.0025                     | 0.0035                     |
|-------------|--------------|------------------------------|------------------------------|------------------------------|------------------------------|------------------------------|
|             | CQL-P        | -500.0$\pm$ 0.0 | -500.0$\pm$ 0.0 | -500.0$\pm$ 0.0 | -500.0$\pm$ 0.0 | -500.0$\pm$ 0.0 |
|             | CQL-A        | -500.0$\pm$ 0.0 | -500.0$\pm$ 0.0 | -500.0$\pm$ 0.0 | -500.0$\pm$ 0.0 | -500.0$\pm$ 0.0 |
|             | PerSim-CQL-5 | -44.7$\pm$ 0.1  | -49.8$\pm$0.0   | -63.2$\pm$0.1   | -500.0$\pm$0.0  | -500.0$\pm$ 0.0 |

### Table 4: CQL + PerSim: CartPole (Random data)
|    | Method     | (2/0.5)                    | (10.0/0.5)                 | (18.0/0.5)                 | (10/0.85)                  | (10/0.15)                  |
|-------------|--------------|------------------------------|------------------------------|------------------------------|------------------------------|------------------------------|
|             | CQL-P        | 39.9$\pm$29.9   | 72.3$\pm$41.7   | 67.8$\pm$ 44.4  | 77.9$\pm$26.2   | 148.7$\pm$ 17.1 |
|             | CQL-A        | 67.4$\pm$49.1   | 9.3$\pm$0.1     | 16.3$\pm$9.9    | 30.6$\pm$26.4   | 7.0 $\pm$1.9    |
|             | PerSim-CQL-5 | 81.8$\pm$3.4    | 198.3$\pm$1.3   | 200.0$\pm$0.0   | 135.5$\pm$11.1  | 190.0$\pm$14.2  |

## Ant Experiments
Further, we now include additional experiments with the Ant environment, which as stated by the reviewer is an environment with high-dimensional state. See table below for reference.

### Table 5: Ant Experiments

| Data | Method       | Agent 1                       | Agent 2                       | Agent 3                       | Agent 4                       | Agent 5                       |
|------|--------------|-------------------------------|-------------------------------|-------------------------------|-------------------------------|-------------------------------|
|    Random  | PerSim    | **114.06**$\pm$31.40 | **121.04**$\pm$26.35 | **129.09**$\pm$6.73  | **106.92**$\pm$30.59 | **102.72**$\pm$27.12 |
|      | Vanilla CaDM | 31.31$\pm$1.88   | 25.11$\pm$10.99  | 27.97$\pm$6.16   | 18.23$\pm$3.97   | 24.81$\pm$10.47  |
|      | BCQ-P        | 52.67$\pm$ 0.27               | 53.12$\pm$0.35                | 53.21$\pm$0.25                | 53.33$\pm$0.46                | 53.24$\pm$0.46                |
|      | BCQ-A        | 35.05$\pm$1.65                | 42.45$\pm$17.09               | 35.26$\pm$9.24                | 23.08$\pm$9.07                | 14.62$\pm$6.38                |


In our final manuscript, we will include the full experiments, including adding MOReL and CQL as baselines, for the HalfCheetah and Ant environments as well. This will take a few weeks to run.


**References:**
1. Kidambi, Rahul, et al. "Morel: Model-based offline reinforcement learning." arXiv preprint arXiv:2005.05951 (2020).
2. A. Kumar, A. Zhou, G. Tucker, and S. Levine. Conservative q-learning for offline reinforcement learning. arXiv preprint arXiv:2006.04779, 2020.

---

### Decision · Program_Chairs · 2021-09-27

**Decision:**

Accept (Poster)

**Comment:**

This paper studies an interesting and challenging offline RL problem, where the agents are heterogeneous and there is only a single trajectory for every agent under a potentially non-optimal policy. The authors propose PerSim, where a personalized simulator is learned for each agent based on the trajectories from all agents, and then a policy is made based on MPC over simulators ensemble.
The technical novelty of the paper lies in the finding that the transition dynamics across agents can be represented as a latent function and thus can be nicely approximated by a low-rank decomposition of the latent functions of agent, state, actions. In general, the contributions of this paper are clear and sufficient.
The majority of reviewers provide acceptance recommendations for this paper. The only reviewer of score 5 raised some questions about the generalization ability and sample complexity of the method, problem setting, and the evaluation, which I think have been well answered with detailed revision plans. Thus I think this paper can be accepted.